

# Exact thermal properties of free-fermionic spin chains

Michał Białończyk[1], Fernando Javier Gómez-Ruiz[2,3] and Adolfo del Campo[4,2,5,6*]

1 Institute of Physics, Jagiellonian University, Łojasiewicza 11, 30-348 Kraków, Poland
2 Donostia International Physics Center, E-20018 San Sebastián, Spain
3 Departamento de Física, Universidad de Los Andes, A.A. 4976, Bogotá, Colombia
4 Department of Physics and Materials Science, University of Luxembourg,
L-1511 Luxembourg, G. D. Luxembourg
5 IKERBASQUE, Basque Foundation for Science, E-48013 Bilbao, Spain
6 Department of Physics, University of Massachusetts, Boston, MA 02125, USA

⋆ adolfo.delcampo@uni.lu

## Abstract

An exact description of integrable spin chains at finite temperature is provided using an elementary algebraic approach in the complete Hilbert space of the system. We focus on spin chain models that admit a description in terms of free fermions, including paradigmatic examples such as the one-dimensional transverse-field quantum Ising and XY models. The exact partition function is derived and compared with the ubiquitous approximation in which only the positive parity sector of the energy spectrum is considered. Errors stemming from this approximation are identified in the neighborhood of the critical point at low temperatures. We further provide the full counting statistics of a wide class of observables at thermal equilibrium and characterize in detail the thermal distribution of the kink number and transverse magnetization in the transverse-field quantum Ising chain.



# 1 Introduction

Quantum many-body spin systems that are exactly solvable and exhibit a quantum phase transition have been key to advance our understanding of critical phenomena in the quantum domain. Among them, the one-dimensional XY model and the closely-related transverse-field quantum Ising model (TFQIM) occupy a unique status, and are paradigmatic test-beds of quantum critical behavior [1–4]. They belongs to a family of models that admit an exact diagonalization by a combination of Jordan-Wigner and Fourier transformations, yielding a formulation of the system in terms of free fermions [5, 6]. These family of quasi-free fermion models include as well the Kitaev spin model in one dimension and on a honeycomb lattice [7], among other examples [1, 2, 4].

Quasi-free fermion models have indeed been instrumental in exploring both equilibrium and nonequilibrium properties. At equilibrium, the study of the ground-state critical behavior was shown to be of relevance to the characterization of the system at finite temperature [8–10]. Out of equilibrium, these models have been used to explore the dynamics following a sudden quench (e.g., of the magnetic field). The study of finite-time quenches was key to establish the validity of the universal Kibble-Zurek mechanism in the quantum domain, and confirm the power-law scaling of the number of kinks by driving the ground-state of a paramagnet across the phase transition [11, 12], as reported in a variety of experiments [13–16]. These results have also been extended to nonlinear quenches [17,18] and inhomogeneous systems [19–23], while their breakdown has been characterized in open systems [24–26]. More recently, it has been shown that signatures of universality are present in the full kink-number distribution and that all cumulants scale as a universal power-law of the quench time [16,27–31]. The universal dynamics of defect formation is not always desirable, and a variety of works have been devoted to circumvent it using diverse control protocols [32–42], beyond the use of nonlinear quenches and inhomogeneous driving. In addition, quasi-free fermion models have been discussed in the context of quantum thermodynamics, as a test-bed to explore work statistics and fluctuation theorems [43–46] and as a working substance in a quantum thermodynamic cycle [47].

Quasi-free fermion models provided an effective description of a variety of condensed-matter systems, where they can be realized with high accuracy in [48]. They are further amenable to quantum simulation with trapped ions [49–53], ultracold gases in optical lattices [54] and superconducting qubits [55]. Digital quantum simulation provides yet another avenue for their study in the laboratory [15, 56–58].

In many applications, it is generally desirable to consider a thermal state and analyze the finite-temperature behavior. For a given observable, full information about the eigenvalue distribution and its cumulants can be extracted from the characteristic function. An ubiquitous approximation in such description exploits the parity symmetry of the TFQIM and XY modes, focusing on the positive-parity subspace, while disregarding the rest of the spectrum [1–4, 44, 59–62]. We refer to it as the positive-parity approximation or PPA for short. The PPA is considered to be accurate in the thermodynamic limit [63], invoked in many works [6, 64]. However, *even in the thermodynamic limit*, an exact treatment requires taking into account parity properly and at finite temperature both subspaces are populated. Katsura derived the exact partition function for a finite-size spin chain in 1962 [63]. Kapitonov and Il'inskii provided an alternative derivation of the closed form expression of the exact partition function using functional integrals over Grassmann variables [65]. More recently, Fei and Quan [45] used group theory methods to calculate the exact partition function and quantum work distribution.

In this manuscript, we first elaborate on these results providing an elementary derivation of the exact partition function based on the structure of the Hilbert space. Using this approach, we next provide exact expressions for the eigenvalue distribution (full counting statistics) of

a wide class of observables at thermal equilibrium. We present step-by-step worked examples deriving the exact moment generating function for important observables: the kink number and transverse magnetization. In addition, we analyze finite-size effects and illustrate discrepancies between results obtained using the PPA for the partition function and the exact partition function for small systems spins. These discrepancies are of direct relevance to typical system sizes in current experimental realizations of spin systems [66, 67]. For convenience of the reader interested in using the final results of a calculation, the corresponding explicit formulas are summarized in boxes that are self-contained and make little or no reference to the rest of the manuscript.

## 2  Full Diagonalization of Spin-$\frac{1}{2}$ XY Model

We consider the anisotropic one-dimensional XY Hamiltonian for spins $1/2$ in a transverse magnetic field $g$. The Hamiltonian reads:

$$\hat{\mathcal{H}}(g,\gamma) = -J\left[\sum_{n=1}^{L}\left(\frac{1+\gamma}{2}\right)\hat{X}_n\hat{X}_{n+1} + \left(\frac{1-\gamma}{2}\right)\hat{Y}_n\hat{Y}_{n+1} + g\hat{Z}_n\right]. \tag{1}$$

Here, $J$ parameterizes the ferromagnetic ($J > 0$) or antiferromagnetic ($J < 0$) exchange interaction between nearest neighbors; we set the energy scale by taking $J = 1$. The dimensionless anisotropic parameter in the XY plane is given by $\gamma > 0$ and $L$ is the number of sites in the chain. For $\gamma = 1$, the Hamiltonian (1) corresponds to the Ising model in a transverse magnetic field, which possesses a $\mathbb{Z}_2$ symmetry. The limit $\gamma = 0$ describes the isotropic XY model. For the anisotropic case $0 < \gamma \leq 1$ the model belongs to the Ising universality class, and its phase diagram is determined by the ratio $\nu = g/J$. When $\nu > 1$, the magnetic field dominates over the nearest-neighbor coupling, polarizing the spins along the $z$ direction. This corresponds to a paramagnetic state, with zero magnetization in the $xy$ plane. By contrast, in the regime $0 \leq \nu < 1$ the ground state of the system corresponds to a ferromagnetic configuration with polarization along the $xy$ plane. These phases are separated by a quantum phase transition (QPT) at the critical point $\nu = 1$. Finally, for the isotropic case $\gamma = 0$, a QPT is observed between gapless ($\nu < 1$) and ferromagnetic ($\nu > 1$) phases.

The operators $\hat{X}_n$, $\hat{Y}_n$, and $\hat{Z}_n$ are matrices of order $2^L$ defined by the relations

$$\begin{aligned}
\hat{X}_n &= \hat{\mathbb{1}}_1 \otimes \ldots \otimes \hat{\mathbb{1}}_{n-1} \otimes \hat{\sigma}_n^x \otimes \hat{\mathbb{1}}_{n+1} \otimes \ldots \otimes \hat{\mathbb{1}}_L, \\
\hat{Y}_n &= \hat{\mathbb{1}}_1 \otimes \ldots \otimes \hat{\mathbb{1}}_{n-1} \otimes \hat{\sigma}_n^y \otimes \hat{\mathbb{1}}_{n+1} \otimes \ldots \otimes \hat{\mathbb{1}}_L, \\
\hat{Z}_n &= \hat{\mathbb{1}}_1 \otimes \ldots \otimes \hat{\mathbb{1}}_{n-1} \otimes \hat{\sigma}_n^z \otimes \hat{\mathbb{1}}_{n+1} \otimes \ldots \otimes \hat{\mathbb{1}}_L.
\end{aligned} \tag{2}$$

Here, $\hat{\sigma}_n^\alpha$ denotes the Pauli operator at site $n$ along the axis $\alpha = x, y, z$, $\hat{\mathbb{1}}_n$ is the identity matrix of order 2 at the site $n$, and periodic boundary conditions are assumed, $\hat{\sigma}_{L+1}^\alpha = \hat{\sigma}_1^\alpha$. A standard way to diagonalize the Hamiltonian in Eq. (1) relies on introducing a new set of Fermionic operators given by

$$\begin{aligned}
\hat{\sigma}_n^x &= \left(\hat{c}_n^\dagger + \hat{c}_n\right)\prod_{m<n}\left(\hat{\mathbb{1}}_m - 2\hat{c}_m^\dagger\hat{c}_m\right), \\
\hat{\sigma}_n^y &= -i\left(\hat{c}_n^\dagger - \hat{c}_n\right)\prod_{m<n}\left(\hat{\mathbb{1}}_m - 2\hat{c}_m^\dagger\hat{c}_m\right), \\
\hat{\sigma}_n^z &= \hat{\mathbb{1}}_n - 2\hat{c}_n^\dagger\hat{c}_n.
\end{aligned} \tag{3}$$

These expressions represent the well-known Jordan-Wigner transformation [68]. Here, $\hat{c}_n$ and $\hat{c}_n^\dagger$ are ladder Fermionic operators at site $n$, which satisfy anti-commutation relations

$\left\{\hat{c}_i,\hat{c}_j^\dagger\right\} = \delta_{i,j}$ and $\left\{\hat{c}_i,\hat{c}_j\right\} = \left\{\hat{c}_i^\dagger,\hat{c}_j^\dagger\right\} = 0$. This is in contrast to the Pauli matrices, which satisfy commutation relations $\left[\hat{\sigma}_n^\dagger,\hat{\sigma}_m^-\right] = \delta_{n,m}\hat{\sigma}_n^z$ and $\left[\hat{\sigma}_n^z,\hat{\sigma}_m^\pm\right] = \pm2\delta_{n,m}\hat{\sigma}_n^\pm$ with $\hat{\sigma}_n^\pm = \hat{\sigma}_n^x \pm i\hat{\sigma}_n^y$. With periodic boundary conditions in the spin representation, the Fermionic operators $\hat{c}_n$ and $\hat{c}_n^\dagger$ satisfy nontrivial boundary conditions

$$\hat{c}_{L+1}^\dagger = (-1)^{\hat{N}}\,\hat{c}_1^\dagger, \qquad\qquad \hat{c}_{L+1} = (-1)^{\hat{N}}\,\hat{c}_1, \qquad (4)$$

where $\hat{N} = \sum_{n=1}^L \hat{c}_n^\dagger\hat{c}_n$ is the Fermionic number operator. By direct substitution of Eq. (3) into Eq. (1), the Hamiltonian can be written as a quadratic form

$$\begin{aligned}
\hat{H}(g,\gamma) = &-\sum_{n=1}^{L-1}\left[\hat{c}_n^\dagger\hat{c}_{n+1} + \hat{c}_{n+1}^\dagger\hat{c}_n + \gamma\left(\hat{c}_n^\dagger\hat{c}_{n+1}^\dagger + \hat{c}_{n+1}\hat{c}_n\right)\right]\\
&+\hat{\Pi}\left[\hat{c}_L^\dagger\hat{c}_1 + \hat{c}_1^\dagger\hat{c}_L + \gamma\left(\hat{c}_L^\dagger\hat{c}_1^\dagger + \hat{c}_1\hat{c}_L\right)\right] - g\sum_{n=1}^L\left(\mathbb{1}_n - 2\hat{c}_n^\dagger\hat{c}_n\right).
\end{aligned} \qquad (5)$$

Here, the parity operator $\hat{\Pi}$ is given by $(-1)^{\hat{N}} = \exp\left(i\pi\hat{N}\right)$ and has eigenvalues $\pm1$. The parity operator anticommutes with the creation $\hat{c}_n^\dagger$ and annihilation $\hat{c}_n$ Fermionic operators, $\left\{(-1)^{\hat{N}},\hat{c}_n^\dagger\right\} = \left\{(-1)^{\hat{N}},\hat{c}_n\right\} = 0$, and therefore, it commutes with any operator bilinear in $\hat{c}_n^\dagger$ and $\hat{c}_n$. The Hamiltonian given by Eq. (5) does not conserve the number of Fermionic excitations. However, it is well-known that the TFQIM has a global $\mathbb{Z}_2$ symmetry and, thus, the parity operator $\hat{\Pi}$ commutes with the Hamiltonian. As a result, the total Hilbert space is split into the direct sum of two $2^{L-1}$ dimensional subspaces of positive $(+1)$ and negative $(-1)$ parity. Using the projectors $\hat{\Pi}^\pm$,

$$\hat{\Pi}^\pm = \frac{1}{2}\left[\hat{\mathbb{1}} \pm (-1)^{\hat{N}}\right], \qquad (6)$$

the Hamiltonian in Eq. (5) is represented in the form

$$\hat{H} = \hat{H}^+\hat{\Pi}^+ + \hat{H}^-\hat{\Pi}^-, \qquad (7)$$

with the reduced Hamiltonians $\hat{H}^\pm$ being given by

$$\hat{H}^\pm(g,\gamma) = -\sum_{n=1}^L\left[\hat{c}_n^\dagger\hat{c}_{n+1} + \hat{c}_{n+1}^\dagger\hat{c}_n + \gamma\left(\hat{c}_n^\dagger\hat{c}_{n+1}^\dagger + \hat{c}_{n+1}\hat{c}_n\right) + g\left(\hat{\mathbb{1}}_n - 2\hat{c}_n^\dagger\hat{c}_n\right)\right]. \qquad (8)$$

A subtle difference between $\hat{H}^+$ and $\hat{H}^-$ is found in the boundary conditions for the Fermion operators. $\hat{H}^+$ obeys antiperiodic boundary conditions ($\hat{c}_{L+1} = -\hat{c}_1$ and $\hat{c}_{L+1}^\dagger = -\hat{c}_1^\dagger$) while $\hat{H}^-$ satisfies periodic boundary conditions ($\hat{c}_{L+1} = \hat{c}_1$ and $\hat{c}_{L+1}^\dagger = \hat{c}_1^\dagger$). The Hamiltonian given by Eq. (8) is quadratic in the Fermionic operators and is thus exactly diagonalizable using Fourier and Bogoliubov transformations [64,69–71]. We expand the operator $\hat{c}_n$ via a Fourier transformation in momentum space,

$$\hat{c}_n = \frac{e^{-i\pi/4}}{\sqrt{L}}\sum_{k\in\mathbf{K}^\pm}\hat{c}_k\exp(ink), \qquad\qquad \hat{c}_n^\dagger = \frac{e^{i\pi/4}}{\sqrt{L}}\sum_{k\in\mathbf{K}^\pm}\hat{c}_k^\dagger\exp(-ink). \qquad (9)$$

The wavevector $k$ takes values in the positive $\left(\mathbf{K}^+\right)$ and negative $\left(\mathbf{K}^-\right)$ parity sectors

$$\mathbf{K}^+ = \left\{k\left|\frac{\pi}{L}(2m-1)\right., \qquad m = -\frac{L}{2}+1,-\frac{L}{2}+2,\ldots,\frac{L}{2}-1,\frac{L}{2}\right\}, \qquad (10)$$

$$\mathbf{K}^- = \left\{k\left|\frac{2\pi}{L}m\right., \qquad m = -\frac{L}{2}+1,-\frac{L}{2}+2,\ldots,\frac{L}{2}-1,\frac{L}{2}\right\}. \qquad (11)$$

We emphasize that Eqs. (10) and (11) are valid for an even and odd number of particles in the chain. In the following analysis, we consider even $L$. In this way, the modes $\mathbf{k} = 0$ and $\mathbf{k} = \pi$ are included in the negative parity sector. For even $L$, we can rewrite conveniently the momentum values as

$$\mathbf{K}^+ = \left\{\pm\frac{\pi}{L}, \pm\frac{3\pi}{L}, \pm\frac{5\pi}{L}, \ldots, \pm\frac{\pi(L-1)}{L}\right\} = \mathbf{k}^+ \cup \{-\mathbf{k}^+\},$$

$$\mathbf{K}^- = \left\{0, \pm\frac{2\pi}{L}, \pm\frac{4\pi}{L}, \ldots, \pm\frac{\pi(L-2)}{L}, \pi\right\} = \mathbf{k}^- \cup \{-\mathbf{k}^-\} \cup \{0, \pi\},$$

with

$$\mathbf{k}^+ = \left\{\frac{\pi}{L}, \frac{3\pi}{L}, \ldots, \frac{\pi(L-1)}{L}\right\}, \qquad \text{and} \qquad \mathbf{k}^- = \left\{\frac{2\pi}{L}, \frac{4\pi}{L}, \ldots, \frac{\pi(L-2)}{L}\right\}. \tag{12}$$

By direct substitution of Eq. (9) into Eq. (8), the reduced Hamiltonians $\hat{H}^+$ and $\hat{H}^-$ are expressed in terms of $\hat{c}_k$ and $\hat{c}_k^\dagger$ as

$$\hat{H}^+(g, \gamma) = \sum_{k \in \mathbf{k}^+} \hat{H}_k(g, \gamma),$$

$$\hat{H}^-(g, \gamma) = \sum_{k \in \mathbf{k}^-} \hat{H}_k(g, \gamma) + \hat{H}_0(g) + \hat{H}_\pi(g), \tag{13}$$

where

$$\hat{H}_k(g, \gamma) = 2\left[(g - \cos(k))\left(\hat{c}_k^\dagger \hat{c}_k - \hat{c}_{-k}\hat{c}_{-k}^\dagger\right) + \gamma \sin(k)\left(\hat{c}_k^\dagger \hat{c}_{-k}^\dagger - \hat{c}_{-k}\hat{c}_k\right)\right],$$

$$\hat{H}_0(g) = (g - 1)\left(\hat{c}_0^\dagger \hat{c}_0 - \hat{c}_0 \hat{c}_0^\dagger\right), \tag{14}$$

$$\hat{H}_\pi(g) = (g + 1)\left(\hat{c}_\pi^\dagger \hat{c}_\pi - \hat{c}_\pi \hat{c}_\pi^\dagger\right).$$

We next make use of a Bogoliubov transformation, and define a new set of fermion operators $\hat{\gamma}_k$ and $\hat{\gamma}_k^\dagger$ given by

$$\hat{\gamma}_k = u_k \hat{c}_k - i v_k \hat{c}_{-k}^\dagger, \qquad\qquad \hat{\gamma}_k^\dagger = u_k \hat{c}_k^\dagger + i v_k \hat{c}_{-k}, \tag{15}$$

where the real numbers $u_k$ and $v_k$ satisfy $u_k = u_{-k}$, $v_k = -v_{-k}$ and $|u_k|^2 + |v_k|^2 = 1$. The canonical anti-commutation relations for the operators $\hat{c}_k$ and $\hat{c}_k^\dagger$ imply that the same relations are also satisfied by $\hat{\gamma}_k$ and $\hat{\gamma}_k^\dagger$, that is, $\{\hat{\gamma}_k, \hat{\gamma}_{k'}^\dagger\} = \delta_{k,k'}$, and $\{\hat{\gamma}_k^\dagger, \hat{\gamma}_{k'}^\dagger\} = \{\hat{\gamma}_k, \hat{\gamma}_{k'}\} = 0$. By direct substitution of the Bogoliubov transformations into Eq. (13), after a some algebra, we obtain

$$\begin{aligned}\hat{H}_k(g, \gamma) = &\, 2\hat{\gamma}_k^\dagger \hat{\gamma}_k \left[u_k^2(\cos(k) - g) + \gamma \sin(k) u_k v_k\right] \\ &+ 2\hat{\gamma}_k \hat{\gamma}_k^\dagger \left[(\cos(k) - g) v_k^2 - \gamma \sin(k) u_k v_k\right] \\ &- i\hat{\gamma}_k \hat{\gamma}_{-k} \left[\gamma \sin(k)\left(u_k^2 - v_k^2\right) + 2(\cos(k) - g) u_k v_k\right] \\ &- i\hat{\gamma}_k^\dagger \hat{\gamma}_{-k}^\dagger \left[\gamma \sin(k)\left(u_k^2 - v_k^2\right) + 2(\cos(k) - g) u_k v_k\right] + g.\end{aligned} \tag{16}$$

The terms proportional to $\gamma_k^\dagger \gamma_{-k}^\dagger$ and $\gamma_k \gamma_{-k}$ should vanish for the Hamiltonian to acquire a diagonal form. Writing $u_k = \cos(\vartheta_k/2)$ and $v_k = \sin(\vartheta_k/2)$, the Bogoliubov angles satisfy

$$\tan(\vartheta_k) = \frac{\gamma \sin(k)}{g - \cos(k)}. \tag{17}$$

For numerical simulations, the last condition can be rewritten as $\gamma \sin(k)\left\{u_k^2 - v_k^2\right\} + 2(\cos(k) - g) u_k v_k = 0$. Finally, the Hamiltonian (13) can be rewritten as a sum of noninteracting terms

$$\hat{H}^+(g, \gamma) = \sum_{k \in \mathbf{k}^+} \epsilon_k(g, \gamma)(\hat{n}_k + \hat{n}_{-k} - 1),$$

$$\hat{H}^-(g, \gamma) = \sum_{k \in \mathbf{k}^-} \epsilon_k(g, \gamma)(\hat{n}_k + \hat{n}_{-k} - 1) + (g - 1)(2\hat{n}_0 - 1) + (g + 1)(2\hat{n}_\pi - 1), \tag{18}$$

with $\hat{n}_k = \hat{\gamma}_k^\dagger \hat{\gamma}_k$ denoting the fermion number operator and $\epsilon_k(g,\gamma) = 2\sqrt{(g-\cos k)^2 + \gamma^2 \sin^2 k}$ being the quasiparticle energy of mode $\mathbf{k} \neq 0, \pi$ per particle.

## 2.1 Mathematical tools for the complete Hilbert space

To simplify the presentation, we focus on the positive-parity subspace in this subsection. However, the methods presented are applicable in the negative-parity sector too. In order to keep the notation clear, we use the following conventions:

- **Hilbert spaces** are denoted by letters in blackboard bold style, for example $\mathbb{H}_k$.

- **Operators** are denoted by letters with a hat, such as $\hat{O}_k$ and $\hat{h}_{k_i}$.

- **Operations** on tensor products of Hilbert spaces are denoted with calligraphic letters $\mathcal{P}$ and $\mathcal{N}$.

To begin with, we note that the positive-parity Hilbert subspace $\mathbb{H}^+$ can be written as the tensor product of subspaces corresponding to each *pair of momenta* ($k$ and $-k$)

$$\mathbb{H}^+ = \bigotimes_{k \in \mathbf{k}^+} \mathbb{H}_k. \tag{19}$$

Each subspace $\mathbb{H}_k$ is the linear span of the vacuum and states involving one and two Fermionic excitations with a given momentum

$$\begin{aligned}
\mathbb{H}_k &= \text{span}\{|0\rangle_k, \hat{c}_k^\dagger \hat{c}_{-k}^\dagger |0\rangle_k, \hat{c}_k^\dagger |0\rangle_k, \hat{c}_{-k}^\dagger |0\rangle_k\} \\
&= \{|00\rangle_k, |11\rangle_k, |10\rangle_k, |01\rangle_k\}, \quad \forall \quad k \in \mathbf{k}^+.
\end{aligned} \tag{20}$$

Here, $|0\rangle_k$ is the vector annihilated by both $\hat{c}_k$ and $\hat{c}_{-k}$. Each of the subspaces can be divided into the sectors with even $\mathbb{H}_k^{(p)}$ and odd $\mathbb{H}_k^{(n)}$ number of excitations

$$\begin{aligned}
\mathbb{H}_k^{(p)} &= \text{span}\{|0\rangle_k, \hat{c}_k^\dagger \hat{c}_{-k}^\dagger |0\rangle_k\} = \{|00\rangle_k, |11\rangle_k\}, \\
\mathbb{H}_k^{(n)} &= \text{span}\{\hat{c}_{-k}^\dagger |0\rangle_k, \hat{c}_k^\dagger |0\rangle_k\} = \{|01\rangle_k, |10\rangle_k\}.
\end{aligned} \tag{21}$$

Note that the dimension of the right hand side of equation (19) is equal to $4^{L/2} = 2^L$, as there are $L/2$ positive momenta and each corresponding subspace is four-dimensional. However, there is an additional condition in the positive-parity subspace: the parity operator $\hat{\Pi}$ has eigenvalue $+1$. Thus, the subspace is only spanned by vectors associated with an even number of quasiparticles. We denote this subspace by $\mathcal{P}(\bigotimes_{k \in \mathbf{k}^+} \mathbb{H}_k)$

$$\mathcal{P} = \mathcal{P}\left(\bigotimes_{k \in \mathbf{k}^+} \mathbb{H}_k\right) = \text{span}\left\{\bigotimes_{k \in \mathbf{k}^+} |i_k j_k\rangle : i_k, j_k \in \{0,1\}, \sum_{k \in \mathbf{k}^+} (i_k + j_k) \text{ is even}\right\}. \tag{22}$$

Similarly, we define the subspace spanned by odd number of quasi-particle excitations and denote it by $\mathcal{N} = \mathcal{N}(\bigotimes_{k \in \mathbf{k}^+} \mathbb{H}_k)$. It is easy to see that both spaces $\mathcal{P}(\bigotimes_{k \in \mathbf{k}^+} \mathbb{H}_k)$ and $\mathcal{N}(\bigotimes_{k \in \mathbf{k}^+} \mathbb{H}_k)$ have dimension $2^{L-1}$ and satisfy

$$\mathbb{H}^+ = \mathcal{P}\left(\bigotimes_{k \in \mathbf{k}^+} \mathbb{H}_k\right) \oplus \mathcal{N}\left(\bigotimes_{k \in \mathbf{k}^+} \mathbb{H}_k\right). \tag{23}$$

For the positive-parity subspace only $\mathcal{P}$ is relevant; vectors in $\mathcal{N}$ have no physical meaning for the system described by the Hamiltonian $\hat{H}^+$. However, the spaces $\mathcal{P}$ and $\mathcal{N}$ (defined for proper momenta) exchange their roles for $\hat{H}^-$; see Eq. (18). These considerations suggest that

to obtain correct results in the positive-parity subspace, it is sufficient to redefine the tensor product to take into account only vectors from $\mathcal{P}$. This can be done for states and observables. Before dealing with observables, we introduce an alternative recursive definition of the spaces $\mathcal{P}$ and $\mathcal{N}$, equivalent to Eq. (22). We shall make use of it in deriving the exact partition function and characteristic functions of observables. We start by defining the subspaces for one momentum, see Eq. (21),

$$\mathcal{P}\left(\mathbb{H}_{k_1}\right) = \mathbb{H}_{k_1}^{(p)}, \quad \mathcal{N}\left(\mathbb{H}_{k_1}\right) = \mathbb{H}_{k_1}^{(n)}. \tag{24}$$

Next, we specify how to construct spaces $\mathcal{P}$ and $\mathcal{N}$ when a mode with momentum $k_{n+1}$ is added:

$$\mathcal{P}\left(\bigotimes_{i=1}^{n+1}\mathbb{H}_{k_i}\right) = \mathcal{P}\left(\bigotimes_{i=1}^{n}\mathbb{H}_{k_i}\right)\otimes \mathbb{H}_{k_{n+1}}^{(p)} \oplus \mathcal{N}\left(\bigotimes_{i=1}^{n}\mathbb{H}_{k_i}\right)\otimes \mathbb{H}_{k_{n+1}}^{(n)}, \; n \geq 1,$$

$$\mathcal{N}\left(\bigotimes_{i=1}^{n+1}\mathbb{H}_{k_i}\right) = \mathcal{N}\left(\bigotimes_{i=1}^{n}\mathbb{H}_{k_i}\right)\otimes \mathbb{H}_{k_{n+1}}^{(p)} \oplus \mathcal{P}\left(\bigotimes_{i=1}^{n}\mathbb{H}_{k_i}\right)\otimes \mathbb{H}_{k_{n+1}}^{(n)}, \; n \geq 1. \tag{25}$$

The intuitive meaning of these equations is that in order to obtain an even number of excitations one has to add an even number of excitations to an even number, or an odd number of excitations to an odd number.

We can extend these definitions for operators and density matrices. We assume that operators $\hat{O}_k$ act independently on each subspace $\mathbb{H}_k$ and each $\hat{O}_k$ can be written as a sum of an even part $\hat{O}_k^{(p)}$ and an odd part $\hat{O}_k^{(n)}$ as

$$\hat{O}_k = \hat{O}_k^{(p)} + \hat{O}_k^{(n)}, \quad \hat{O}_k^{(p)}\Big|_{\mathbb{H}_k^{(n)}} = 0, \quad \hat{O}_k^{(n)}\Big|_{\mathbb{H}_k^{(p)}} = 0. \tag{26}$$

The operators $\hat{O}_k^{(p)}$ and $\hat{O}_k^{(n)}$ act on the total space $\mathbb{H}_k$, but have a $2 \times 2$ zero block $0_2$ in the respective subspace. The proper restrictions of the tensor product of operators $\hat{O}_k$ can be defined in a similar way as in Eqs. (24) and (25) for $\mathcal{P}\left(\hat{O}_{k_1}\right) = \hat{O}_{k_1}^{(p)}$ and $\mathcal{N}\left(\hat{O}_{k_1}\right) = \hat{O}_{k_1}^{(n)}$, and are given by

$$\mathcal{P}\left(\bigotimes_{i=1}^{n+1}\hat{O}_{k_i}\right) = \mathcal{P}\left(\bigotimes_{i=1}^{n}\hat{O}_{k_i}\right)\otimes \hat{O}_{k_{n+1}}^{(p)} + \mathcal{N}\left(\bigotimes_{i=1}^{n}\hat{O}_{k_i}\right)\otimes \hat{O}_{k_{n+1}}^{(n)}, \; n \geq 1,$$

$$\mathcal{N}\left(\bigotimes_{i=1}^{n+1}\hat{O}_{k_i}\right) = \mathcal{N}\left(\bigotimes_{i=1}^{n}\hat{O}_{k_i}\right)\otimes \hat{O}_{k_{n+1}}^{(p)} + \mathcal{P}\left(\bigotimes_{i=1}^{n}\hat{O}_{k_i}\right)\otimes \hat{O}_{k_{n+1}}^{(n)}, \; n \geq 1. \tag{27}$$

---

**Example 2.1: Even and odd parity parts of the Hamiltonian**

For $\hat{H}_k$ given by Eq. (14), note that for a each mode $k_n$ the Hamiltonian can be rewritten as

$$\hat{H}_k = \mathcal{P}\left(\hat{\mathbb{1}}_{k_1} \otimes \hat{\mathbb{1}}_{k_2} \otimes \ldots \otimes \hat{h}_{k_n} \otimes \ldots \otimes \hat{\mathbb{1}}_{k_{L/2}}\right),$$

where, in the basis $\{|00\rangle_k, |11\rangle_k, |01\rangle_k, |10\rangle_k\}$,

$$\hat{h}_{k_n} = 2\begin{pmatrix} \cos(k_n) - g & \gamma\sin(k_n) & 0 & 0 \\ \gamma\sin(k_n) & g - \cos(k_n) & 0 & 0 \\ 0 & 0 & 0 & 0 \\ 0 & 0 & 0 & 0 \end{pmatrix}.$$

Here, $\hat{h}_{k_n}^{(n)}$ is $4 \times 4$ zero matrix (with no odd part), and $\hat{h}_{k_n}^{(p)} = \hat{h}_{k_n}$.

As the odd part of Hamiltonian is zero, the description using ordinary tensor products instead of over $\mathcal{P}$ is valid for pure states. However, the canonical thermal Gibbs state has a non-vanishing odd-parity contribution:

---

**Example 2.2: Even and odd-parity contributions to the exact Gibbs state**

Consider the part of the thermal Gibbs state corresponding to momentum $k$:

$$\hat{\rho}_k = \exp\left(-\beta \hat{h}_k\right). \tag{28}$$

Using the expression for $\hat{h}_k$ in the the basis $\{|00\rangle_k, |11\rangle_k, |01\rangle_k, |10\rangle_k\}$,

$$\hat{\rho}_k = \exp\left[-2\beta \begin{pmatrix} \cos(k)-g & \gamma\sin(k) \\ \gamma\sin(k) & g-\cos(k) \end{pmatrix}\right] \oplus \mathbb{I}_2. \tag{29}$$

Therefore, the even and odd parts read:

$$\hat{\rho}_k^{(p)} = \exp\left[-2\beta \begin{pmatrix} \cos(k)-g & \gamma\sin(k) \\ \gamma\sin(k) & g-\cos(k) \end{pmatrix}\right] \oplus 0_2, \quad \hat{\rho}_k^{(n)} = 0_2 \oplus \mathbb{I}_2. \tag{30}$$

Using the fact that $\hat{h}_k$ has eigenvalues $\pm\epsilon_k$, we have:

$$\mathrm{Tr}\left(\hat{\rho}_k^{(p)}\right) = 2\cosh\left(\beta\epsilon_k(g,\gamma)\right), \quad \mathrm{Tr}\left(\rho_k^{(n)}\right) = 2. \tag{31}$$

---

Next, we state three propositions helpful in calculating the complete and exact expression of the partition function and the full counting statistics of observables:

---

**Proposition 2.3: Identities for product of operators**

Consider two operators $\hat{O}_k$ and $\hat{R}_k$ acting independently on each subspace $\mathbb{H}_k$. Then, the following identities are true for operator multiplication

$$\mathcal{P}\left(\bigotimes_{i=1}^{n} \hat{O}_{k_i}\right) \mathcal{P}\left(\bigotimes_{i=1}^{n} \hat{R}_{k_i}\right) = \mathcal{P}\left(\bigotimes_{i=1}^{n} \hat{O}_{k_i}\hat{R}_{k_i}\right),$$
$$\mathcal{N}\left(\bigotimes_{i=1}^{n} \hat{O}_{k_i}\right) \mathcal{N}\left(\bigotimes_{i=1}^{n} \hat{R}_{k_i}\right) = \mathcal{N}\left(\bigotimes_{i=1}^{n} \hat{O}_{k_i}\hat{R}_{k_i}\right). \tag{32}$$

---

The following proposition is useful in calculations involving Gibbs states and time-evolutions:

---

**Proposition 2.4: Identities for exponentials of operators**

For every set of operators $O_k$ acting on the subspace $\mathbb{H}_k$, the following identities for exponents of operators hold:

$$\exp\left[\mathcal{P}\left(\bigotimes_{i=1}^{n} \hat{O}_{k_i}\right)\right] = \mathcal{P}\left(\bigotimes_{i=1}^{n} \exp\left(\hat{O}_{k_i}\right)\right),$$
$$\exp\left[\mathcal{N}\left(\bigotimes_{i=1}^{n} \hat{O}_{k_i}\right)\right] = \mathcal{N}\left(\bigotimes_{i=1}^{n} \exp\left(\hat{O}_{k_i}\right)\right). \tag{33}$$

---

Lastly, the use of traces turns out to be essential to determine expectation values of observables, and, more generally, their full counting statistics:

**Proposition 2.5: Trace identities**

Consider operators $\hat{O}_k$ that act independently on each subspace $\mathbb{H}_k$. Then, the traces of the restricted tensor products can be expressed as follows,

$$\text{tr}\left[\mathcal{P}\left(\bigotimes_{i=1}^{n}\hat{O}_{k_i}\right)\right] = \frac{1}{2}\left(\prod_{i=1}^{n}\text{tr}\left(\hat{O}_{k_i}\right) + \prod_{i=1}^{n}\left(\text{tr}\left(\hat{O}_{k_i}^{(p)}\right) - \text{tr}\left(\hat{O}_{k_i}^{(n)}\right)\right)\right),$$

$$\text{tr}\left[\mathcal{N}\left(\bigotimes_{i=1}^{n}\hat{O}_{k_i}\right)\right] = \frac{1}{2}\left(\prod_{i=1}^{n}\text{tr}\left(\hat{O}_{k_i}\right) - \prod_{i=1}^{n}\left(\text{tr}\left(\hat{O}_{k_i}^{(p)}\right) - \text{tr}\left(\hat{O}_{k_i}^{(n)}\right)\right)\right). \tag{34}$$

We present a proof ot Eq. (34) in the Appendix A.

*Negative-parity subspace.* In the negative-parity subspace, all formulas derived for the positive-parity subspace remain valid. In particular, for all momenta $k \neq 0, \pi$ expressions from examples 2.1, 2.2 apply. The only difference is that one has to treat carefully the parts of the Hilbert space associated with momenta $0$ and $\pi$. They are spanned by the following bases:

$$\mathbb{H}_0 = \text{span}\{|0\rangle_0, \hat{c}_0^\dagger|0\rangle_0\},$$

$$\mathbb{H}_\pi = \text{span}\{|0\rangle_\pi, \hat{c}_\pi^\dagger|0\rangle_\pi\}. \tag{35}$$

As a result, matrices describing the Hamiltonian and Gibbs state are $2 \times 2$ instead of $4 \times 4$. In the following example we give formulas for the even- and odd-parity parts of the Gibbs state in modes $k = 0, \pi$:

**Example 2.6: Even- and odd-parity parts of the exact Gibbs state for $0, \pi$ momenta**

Using equation (16), the explicit form of the Gibbs state of the modes with momenta $0, \pi$, in the bases $\{|0\rangle_0, \hat{c}_0^\dagger|0\rangle_0\}, \{|0\rangle_\pi, \hat{c}_\pi^\dagger|0\rangle_\pi\}$, are respectively given by

$$\hat{\rho}_0 = \begin{pmatrix} e^{-\beta(g-1)} & 0 \\ 0 & e^{\beta(g-1)} \end{pmatrix}, \quad \hat{\rho}_\pi = \begin{pmatrix} e^{-\beta(g+1)} & 0 \\ 0 & e^{\beta(g+1)} \end{pmatrix}. \tag{36}$$

Thus, the corresponding even- and odd-parity parts read

$$\hat{\rho}_0^{(p)} = \begin{pmatrix} e^{-\beta(g-1)} & 0 \\ 0 & 0 \end{pmatrix}, \qquad \hat{\rho}_\pi^{(p)} = \begin{pmatrix} e^{-\beta(g+1)} & 0 \\ 0 & 0 \end{pmatrix}, \tag{37a}$$

$$\hat{\rho}_0^{(n)} = \begin{pmatrix} 0 & 0 \\ 0 & e^{\beta(g-1)} \end{pmatrix}, \qquad \hat{\rho}_\pi^{(n)} = \begin{pmatrix} 0 & 0 \\ 0 & e^{\beta(g+1)} \end{pmatrix}. \tag{37b}$$

In closing this section, we point out that when $L$ is odd, the momenta $0$ and $\pi$ appear in the positive-parity subspace; the general formulas (24) and (26) are always valid.

# 3 The Canonical Partition Function

The partition function is a fundamental object in statistical mechanics from which all equilibrium thermal properties of a system can be derived. It further facilitates the study of critical phenomena through the study of its zeroes in the complex plane, know as Lee-Yang zeros [72].

For its study, we consider a linear spin$-1/2$ chain described by Eq. (1). The system is prepared in a canonical thermal Gibbs state at finite inverse temperature $\beta$ and characterized

by the initial density operator

$$\hat{\rho}_{\text{Gibbs}}(\beta, g, \gamma) = \frac{\exp\left(-\beta \hat{H}(g, \gamma)\right)}{Z(\beta, g, \gamma)}, \tag{38}$$

where $Z(\beta, g, \gamma)$ is the canonical partition function given by

$$Z(\beta, g, \gamma) = \text{tr}\left[\exp\left(-\beta \hat{H}(g, \gamma)\right)\right]. \tag{39}$$

In a Gibbs state, the system is in a mixture of positive- and negative-parity states and both subspaces should be taken into account. To this end, we consider the operator $\hat{\rho} = \exp\left(-\beta \hat{H}\right)$, where $\hat{H}$ is given by Eq. (1). According to the exact diagonalization (see Sec. 2), the total Hamiltonian can be mapped to a set of independent mode operators in each parity sector. For fixed even $L$, the operator $\hat{\rho}$ is given by

$$\hat{\rho} = \exp\left[-\beta\left(\hat{H}^+ \hat{\Pi}^+ + \hat{H}^- \hat{\Pi}^-\right)\right] = \hat{\rho}^+ \oplus \hat{\rho}^-, \tag{40}$$

where

$$\hat{\rho}^+ = \mathcal{P}\left(\bigotimes_{k \in \mathbf{k}^+} \hat{\rho}_k\right), \qquad \hat{\rho}^- = \mathcal{N}\left(\bigotimes_{k \in \mathbf{k}^-} \hat{\rho}_k \otimes \hat{\rho}_0 \otimes \hat{\rho}_\pi\right), \tag{41}$$

and $\hat{\rho}_k$ are defined in Examples 2.2, with the sets $\mathbf{k}^+$ and $\mathbf{k}^-$ given in Eq. (12). For these operators the corresponding reduced partition functions are

$$Z^+(\beta, g, \gamma) = \text{tr}\left[\mathcal{P}\left(\bigotimes_{k \in \mathbf{k}^+} \hat{\rho}_k\right)\right], \quad \text{and} \quad Z^-(\beta, g, \gamma) = \text{tr}\left[\mathcal{N}\left(\bigotimes_{k \in \mathbf{k}^-} \hat{\rho}_k \otimes \hat{\rho}_0 \otimes \hat{\rho}_\pi\right)\right]. \tag{42}$$

For simplicity, we calculate $Z^+$ and $Z^-$ separately, and focus on $Z^+$ first. Using the formulas from Example 2.2, one finds

$$\text{tr}(\hat{\rho}_k) = 2\cosh(\beta \epsilon_k) + 2 = 4\cosh^2\left(\frac{\beta \epsilon_k}{2}\right),$$
$$\text{tr}\left(\hat{\rho}_k^{(p)}\right) - \text{tr}\left(\hat{\rho}_k^{(n)}\right) = 2\cosh(\beta \epsilon_k) - 2 = 4\sinh^2\left(\frac{\beta \epsilon_k}{2}\right). \tag{43}$$

Making use of the first identity in (34), we obtain an expression for canonical partition function in the positive-parity sector

$$Z^+(\beta, g, \gamma) = \frac{1}{2}\left(\prod_{k \in \mathbf{k}^+} 2^2 \cosh^2\left(\frac{\beta}{2}\epsilon_k(g, \gamma)\right) + \prod_{k \in \mathbf{k}^+} 2^2 \sinh^2\left(\frac{\beta}{2}\epsilon_k(g, \gamma)\right)\right). \tag{44}$$

The computation of the negative-parity part of the partition function proceeds in the same way; we use the second of the trace identities (34) and the expressions from the example 2.1 to find

$$Z^-(\beta, g, \gamma) = \frac{1}{2}\left(2^2 \cosh(\beta(g+1))\cosh(\beta(g-1))\prod_{k \in \mathbf{k}^-} 2^2 \cosh^2\left(\frac{\beta}{2}\epsilon_k(g, \gamma)\right)\right.$$
$$\left. - 2^2 \sinh(\beta(g+1))\sinh(\beta(g-1))\prod_{k \in \mathbf{k}^-} 2^2 \sinh^2\left(\frac{\beta}{2}\epsilon_k(g, \gamma)\right)\right). \tag{45}$$

Using (40), the exact partition is the sum of contributions of positive and negative parity: $Z(\beta, g, \lambda) = Z^+(\beta, g, \gamma) + Z^-(\beta, g, \gamma)$. To sum up, one can rewrite exact partition function in closed-form.

**Summary 3.1: Exact partition function for spin-$\frac{1}{2}$ XY model**

$$Z(\beta, g, \gamma) = \frac{1}{2} \left( \prod_{k \in \mathbf{K}^+} 2\cosh\left(\frac{\beta}{2}\epsilon_k(g,\gamma)\right) + \prod_{k \in \mathbf{K}^+} 2\sinh\left(\frac{\beta}{2}\epsilon_k(g,\gamma)\right) \right.$$
$$\left. + \prod_{k \in \mathbf{K}^-} 2\cosh\left(\frac{\beta}{2}\epsilon_k(g,\gamma)\right) - \prod_{k \in \mathbf{K}^-} 2\sinh\left(\frac{\beta}{2}\epsilon_k(g,\gamma)\right) \right), \tag{46}$$

where

$$\epsilon_k(g,\gamma) = 2\sqrt{(g - \cos(k))^2 + (\gamma \sin(k))^2}, \quad \epsilon_{k=0} = 2(g-1), \quad \epsilon_{k=\pi} = 2(g+1). \tag{47}$$

In this expression the products run over *all momenta*, not only those with non-negative values. In general, the total partition function can be represented as the sum of four contributions,

$$Z(\beta, g, \gamma) = \frac{1}{2} \left[ Z_F^+(\beta, g, \gamma) + Z_F^-(\beta, g, \gamma) + Z_B^+(\beta, g, \gamma) - Z_B^-(\beta, g, \gamma) \right], \tag{48}$$

where $Z_F^\pm(\beta, g, \gamma) = \prod_{k \in \mathbf{K}^\pm} 2\cosh(\beta \epsilon_k(g,\gamma)/2)$ and $Z_B^\pm(\beta, g, \gamma) = \prod_{k \in \mathbf{K}^\pm} 2\sinh(\beta \epsilon_k(g,\gamma)/2)$ are the "Fermionic" and "boundary" contributions. The first term, which takes only into account Fermionic and positive-parity contribution is the only term considered in the PPA, widely used in the literature as the correct approximation in the limit $N \to \infty$ [1, 44, 59, 60, 62, 64]

**Summary 3.2: PPA partition function**

$$Z_{\text{PPA}}(\beta, g, \gamma) = Z_F^+(\beta, g, \gamma) = \prod_{k \in \mathbf{K}^+} 2\cosh\left(\frac{\beta}{2}\epsilon_k(g,\gamma)\right). \tag{49}$$

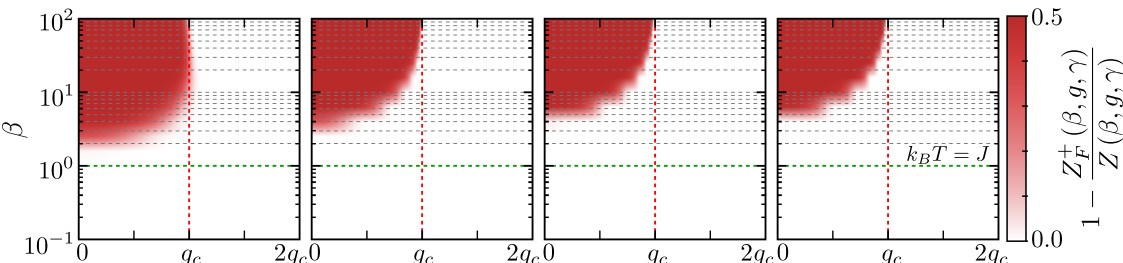

Figure 1: **Comparison of the exact and PPA canonical partition functions.** The ratio between the total partition function 3.1 and the PPA Eq. (49) is shown in the $\beta$-$g$ plane for finite system size $L = 50, 100, 5000, 10000$, increasing from left to right (anisotropic parameter $\gamma = 1$). Significant differences appear close to the critical point $g = g_c = 1$, with the magnitude of $Z_F^+(\beta, g, \gamma)$ deviating by 50% from the exact partition function. The paramagnetic phase is correctly reproduced by the simplified approximation $g > g_c$ while errors in the partition function are shown in red in the ferromagnetic phase at low temperatures.

In the isotropic case with $\gamma = 0$, the exact partition function admits a more compact expression [73] but this limit lies outside the Ising universality class, our primary focus. The complete expression for the partition function 3.1 was first derived with the aid of creation and annihilation operators by Katsura [63]. An alternative approach has been reported using Grassmann variables, without a numerical characterization [65]; see as well [45].

It is thus natural to analyze the extent to which the PPA $Z_F^+(\beta, g, \gamma)$ provides a valid approximation to the exact partition function.

Fig. 1 shows the difference between the ratio $Z_F^+(\beta)/Z(\beta)$ as a function of the inverse of temperature and the magnetic field. The error is negligible away from criticality and at high temperatures. However, prominent discrepancies between the exact partition function 48 and the ubiquitously-used PPA (49) are manifested in the neighborhood of the critical point in the regime of low-temperatures, which is often times the regime studied and of interest. Indeed, in this region errors reach sufficiently large values such that $Z_F^+(\beta, g, \gamma) \approx 0.5\, Z(\beta, g, \gamma)$.

One can provide a simple and intuitive explanation of the magnitude of this discrepancy by considering the structure of the spectrum. The complete spectrum consists of two disjoint "ladders" of levels, spanning the positive-parity and negative-parity subspaces. In the following analysis we denote by $E_g^\alpha$ and $|g^\alpha\rangle$ the lowest energy level and the corresponding eigenstate in the subspace of parity $\alpha = \pm$. The diagonalization procedure of the Ising model yields explicit formulas for these eigenvalues. For even number of spins [74]

$$
\begin{aligned}
E_g^+ &= -\sum_{k \in \mathbf{k}^+} \epsilon_k\,, \\
E_g^- &= -\sum_{k \in \mathbf{k}^-} \epsilon_k - 2\,.
\end{aligned}
\tag{50}
$$

The corresponding eigenstates read

$$
\begin{aligned}
|g^+\rangle &= \prod_{k \in \mathbf{k}^+} (\cos(\vartheta_k/2) - \sin(\vartheta_k/2)\hat{c}_k^\dagger \hat{c}_{-k}^\dagger)|\text{vac}\rangle\,, \\
|g^-\rangle &= c_0^\dagger \prod_{k \in \mathbf{k}^-} (\cos(\vartheta_k/2) - \sin(\vartheta_k/2)\hat{c}_k^\dagger \hat{c}_{-k}^\dagger)|\text{vac}\rangle\,,
\end{aligned}
\tag{51}
$$

where $|\text{vac}\rangle$ is annihilated by all $\hat{c}_k$ for $k \in \mathbf{K}^+ \cup \mathbf{K}^-$ (including 0 and $\pi$ modes). In what follows, we restrict ourselves to the TFQIM ($\gamma = 1$). In the TFQIM with even number of spins $L$, the true ground state always lies in the positive-parity subspace (this is not necessary true in the XY model, see [75]). The energy gap $\delta(g)$ between these two lowest energy states plays a crucial role. We recall its asymptotic behavior [74]

$$
\begin{aligned}
\delta(0 < g < 1) &= \mathcal{O}[\sim \exp(-L/\xi(g))]\,, \\
\delta(g = 1) &= 2\tan\left[\frac{\pi}{4L}\right] \approx \frac{\pi}{2L}\,, \\
\delta(g > 1) &= 2g - 2 + \mathcal{O}\left(g^{-L}\right)\,,
\end{aligned}
\tag{52}
$$

where $\xi(g)$ denotes the correlation length. In the low temperature regime, the Gibbs state is effectively spanned by the two lowest energy states, $|g\rangle^+$ and $|g\rangle^-$. In this truncation, the partition function and Gibbs state read

$$
Z_{\text{approx}}(\beta, g) = e^{-\beta E_g^+} + e^{-\beta E_g^-}\,,
\tag{53}
$$

$$
\rho_{\text{Gibbs}}(\beta, g) \approx \frac{1}{Z_{\text{approx}}(\beta, g)}\left(e^{-\beta E_g^+}|g^+\rangle\langle g^+| + e^{-\beta E_g^-}|g^-\rangle\langle g^-|\right)\,.
\tag{54}
$$

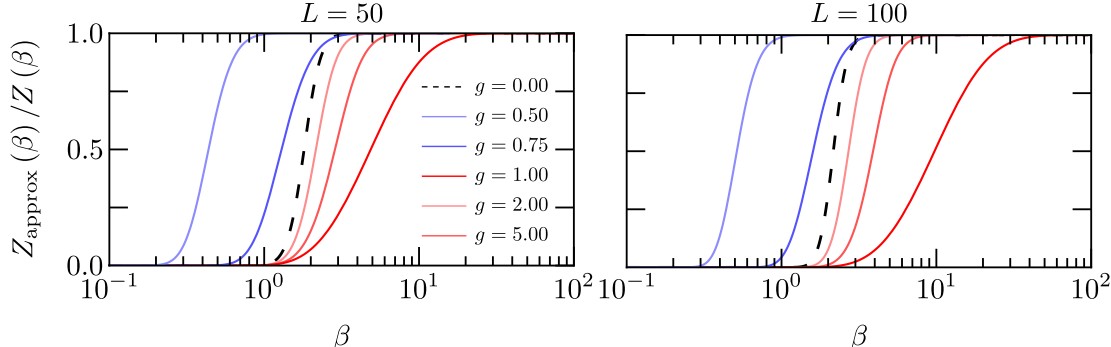

Figure 2: Ratio between the low-temperature approximation and exact partition functions as a function of the inverse temperature. The accuracy of the two-level approximation (53) is considered for different values of the transverse magnetic field $g$ and two different system sizes. As the energy gaps $\delta(g)$ and $\Delta(g)$ in the neighbourhood of $g_c = 1$ are comparable, a lower temperature is required to obtain a desired level of accuracy. For given $\beta$, the accuracy decreases with increasing system size.

This low-temperature two-level approximation relies on (51) and disregards the contribution from higher excited states, that are energetically separated from $|g^+\rangle$ and $|g^-\rangle$. The energy gap to the next excited state can be calculated as the energy of a single-particle excitation in the positive-parity subspace, which sufficiently far from the critical point is estimated by

$$\Delta(g) = 4\sqrt{g^2 - 2g\cos\left(\frac{\pi}{L}\right) + 1} = 4|g - 1| + \mathcal{O}\left(\frac{1}{L^2}\right), \quad g > 0,\ g \neq 1, \qquad (55)$$

while at the critical point, this gap behaves as

$$\Delta(g = 1) \approx \frac{4\pi}{L}. \qquad (56)$$

In the ferromagnetic phase, the first excited state is separated from the ground state by an exponentially vanishing gap and the second excited state lies far away from both of them. Therefore, the correction from high-energy states is negligible in the low temperature limit $\beta\Delta(g) \gg 1$. Similarly, in the paramagnetic phase, the ground state is energetically separated from all the excited states. At the critical point the two lowest excited states are separated from the ground state by a comparable gap,

$$\frac{\Delta(g = 1)}{\delta(g = 1)} \xrightarrow[L \to \infty]{} \frac{1}{8}. \qquad (57)$$

However, for large $\beta$ the error is very small. The accuracy of the the two-level approximation for different phases is shown in Fig. 2. The validity of this approximation (53) explains the magnitude of the errors between the exact and the PPA partition functions shown in Fig. 1. For $g < 1$, the simplified partition function takes into account only the ground state $|g^+\rangle$ and can be approximated by $e^{-\beta E_g^+}$, while the complete partition function is approximately

$$Z_{\text{approx}}(\beta, g) \approx e^{-\beta E_g^+} + e^{-\beta E_g^-} \approx 2e^{-\beta E_g^+}. \qquad (58)$$

This explains the observed error of about 50% between the exact and PPA partition functions.

# 4 Full Counting Statistics in Integrable Spin Chains

The characterization of a given observable in a quantum system generally relies on the study of its expectation value. To determine it, experiments often collect a number of measurements, and build a histogram, from which the eigenvalue distribution is estimated. The full counting statistics of an observable focuses on the complete eigenvalue distribution. Its study has proved useful in a wide variety of applications and alternative methods for its measurement have been put forward [76]. A prominent example concerns the counting statistics of the number of fermions (electrons) traversing a point contact in a wire, that is described by the Levitov-Lesovik formula [77–79]. Distributions of other observables such as the total energy play a key role in quantum chaos [80] and the statistics of related positive-operator valued measures (POVMs, such as work) are at the core of fluctuation theorems in quantum thermodynamics [81]. In the context of spin chains, the distribution of the order parameter has long been recognized as a probe for criticality and turbulence [82–90]. Further, the study of the full counting statistics of quasiparticles and topological defects has been key to uncover universal dynamics of phase transitions beyond the paradigmatic Kibble-Zurek mechanism [16, 27–30, 91].

The full counting statistics is characterized by the probability $P(\omega)$ to obtain the eigenvalue $\omega$ of a general operator $\hat{W}$. It is defined as the expectation value

$$P(\omega) = \left\langle \delta\left(\hat{W} - \omega\right) \right\rangle, \tag{59}$$

where the $\delta$ function is to be interpreted as a Kronecker or Dirac delta function, depending on whether the spectrum of $\hat{W}$ is point-wise or continuous. The angular bracket denotes the quantum expectation value with respect to a general state characterized by a density matrix $\hat{\rho}$. We introduce the Fourier transform representation

$$P(\omega) = \frac{1}{2\pi} \int_{-\infty}^{\infty} d\theta \tilde{P}(\theta) \exp(-i\theta\omega), \tag{60}$$

where $\tilde{P}(\theta)$ is the characteristic function given by

$$\tilde{P}(\theta) = \text{tr}\left[\hat{\rho} \exp\left(i\theta\hat{W}\right)\right]. \tag{61}$$

In cases such as the kink number and the transverse magnetization, the eigenvalues are integers $\omega \in \mathbb{Z}$ and the range of the integral can be restricted from $-\pi$ to $\pi$. The characteristic function is also known as the moment generating function, as it allows to directly compute the mean value and higher-order moments of a given observable $\hat{W}$ according to

$$\langle \hat{W}^m \rangle = \frac{1}{i^m} \frac{d^m}{d\theta^m} \tilde{P}(\theta) \bigg|_{\theta=0}. \tag{62}$$

Further, its logarithm is the cumulant generating function used to derive the cumulants of the distribution through the identity

$$\kappa_m = (-i)^m \frac{d^m}{d\theta^m} \ln \tilde{P}(\theta) \bigg|_{\theta=0}. \tag{63}$$

The first cumulant $\kappa_1$ is just the mean value, $\kappa_2$ is the variance, and $\kappa_3$ coincides with the third central moment. Cumulants are useful in characterizing fluctuations in a quantum system. For example, since the only distribution with finite $\kappa_1, \kappa_2 \neq 0$ and vanishing $\kappa_m = 0$ for $m > 2$ is the Gaussian distribution, higher cumulants quantify non-normal features of the distribution of interest, e.g., an eigenvalue distribution.

We next derive the general form of characteristic function for a wide class $\mathcal{W}$ of observables. This class is defined by the property that any operator $\hat{W} \in \mathcal{W}$, in each parity subspace, can be written in the form

$$\hat{W} = \sum_k \hat{W}_k, \tag{64}$$

where

$$\hat{W}_k = \hat{\Psi}_k^\dagger \hat{w}_k \hat{\Psi}_k, \quad \hat{\Psi}^\dagger = \left( \hat{c}_{-k}, \, \hat{c}_k^\dagger, \, \hat{c}_k, \, \hat{c}_{-k}^\dagger \right) \tag{65}$$

and the matrix $\hat{w}_k$ has the block-diagonal form

$$\hat{w}_k = \begin{pmatrix} \hat{w}_k^{(1)} & 0 \\ 0 & \hat{w}_k^{(2)} \end{pmatrix}. \tag{66}$$

Here, $\hat{w}_k^{(1)}$ and $\hat{w}_k^{(2)}$ are $2 \times 2$ are matrices for momenta different from $0, \pi$ and $1 \times 1$ matrices for $0, \pi$ momenta. We point out that the notation in equations (65, 66) is compatible with matrix expressions from Examples 2.1, 2.2, written in the basis $\{|00\rangle_k, |11\rangle_k, |01\rangle_k, |10\rangle_k\}$. When off-diagonal blocks vanish, the operator $\hat{W}_k$ can be written as a quadratic form in Fermionic operators. However, there are relevant observables which have components linear in Fermionic operators. For example, the longitudinal magnetizations $X_i$ or $Y_i$ do not belong to the class $\mathcal{W}$ as these observables mix the subspaces with different parities. This leads to severe difficulties. Namely, one has to switch between fermionic operators defined for different sets of momenta. For example, for a given $k \in \mathbf{K}^-$ one has to perform inverse Fourier transform to express $c_k$ as a linear combination of fermionic operators in space domain, and then apply Fourier transform with $\mathbf{K}^+$ as a set of momenta. This will cause, that subspaces with different momenta will be all intertwined, contrary to the basic feature exploited in the systems with periodic boundary conditions - that one can perform calculations independently for every momentum. The treatment of such operators is thus beyond the scope of this paper. The systematic treatment of longitudinal magnetization in zero temperature for Ising model with PBC was conducted in [92], with further extensions including observables involving three fermionic operators in [89]. For other approaches, see for example [93] (open boundary conditions) or [61] (exploiting methods of field theory).

In the following we present the detailed procedure for computing characteristic function $\tilde{P}(\theta)$ of a given observable $\hat{W}$ in the class $\mathcal{W}$.

1. First, we fix the state $\hat{\rho}$ to be the thermal-equilibrium Gibbs state, $\hat{\rho} = \hat{\rho}_{\text{Gibbs}}$ given by equation (38). Then, using formulas from Section 2 we can diagonalize the even-parity part of $\hat{\rho}_k$

$$\exp\left[ -2\beta \begin{pmatrix} \cos(k) - g & \gamma \sin(k) \\ \gamma \sin(k) & g - \cos(k) \end{pmatrix} \right] = \hat{S}_k^\dagger \, \text{diag}\left( e^{-\beta \epsilon_k(g,\gamma)}, \, e^{\beta \epsilon_k(g,\gamma)} \right) \hat{S}_k, \tag{67}$$

where

$$\hat{S}_k = \begin{pmatrix} \cos\left(\frac{\vartheta_k}{2}\right) & \sin\left(\frac{\vartheta_k}{2}\right) \\ \sin\left(\frac{\vartheta_k}{2}\right) & -\cos\left(\frac{\vartheta_k}{2}\right) \end{pmatrix} \tag{68}$$

and the angle $\vartheta_k$ satisfies

$$\cos(\vartheta_k) = \frac{2(\cos(k) - g)}{\epsilon_k(g,\gamma)}, \quad \sin(\vartheta_k) = \frac{2\gamma \sin(k)}{\epsilon_k(g,\gamma)}. \tag{69}$$

2. As in the case of the partition function, it is convenient to separate in the full characteristic function the contributions of positive and negative parity:

$$\tilde{P}(\theta) = \frac{1}{Z(\beta, g, \gamma)} \left( \tilde{P}^+(\theta) + \tilde{P}^-(\theta) \right). \tag{70}$$

Using Propositions 2.3 and 2.4, we aim at calculating

$$
\tilde{P}^+(\theta) = \text{tr}\left[\mathcal{P}\left(\bigotimes_{k\in\mathbf{k}^+}\hat{\rho}_k\exp\left(i\theta\hat{w}_k\right)\right)\right], \quad \tilde{P}^-(\theta) = \text{tr}\left[\mathcal{N}\left(\bigotimes_{k\in\mathbf{k}^-}\hat{\rho}_k\exp\left(i\theta\hat{w}_k\right)\right)\right]. \quad (71)
$$

Next, we define the matrix

$$
\hat{\sigma}_k = \hat{S}_k\exp\left(i\theta\hat{w}_k^{(1)}\right)\hat{S}_k^\dagger. \quad (72)
$$

Denoting the eigenvalues of $\hat{w}_k^{(2)}$ by $\mu_k$ and $\lambda_k$ we find

$$
\text{tr}[\hat{\rho}_k\exp(i\theta\hat{w}_k)] = \hat{\sigma}_k^{11}\,e^{-\beta\epsilon_k(g,\gamma)} + \hat{\sigma}_k^{22}\,e^{\beta\epsilon_k(g,\gamma)} + e^{i\theta\mu_k} + e^{i\theta\lambda_k},
$$
$$
\text{tr}\left[\hat{\rho}_k^{(p)}\exp\left(i\theta\hat{w}_k^{(p)}\right)\right] - \text{tr}\left[\hat{\rho}_k^{(n)}\exp\left(i\theta\hat{w}_k^{(n)}\right)\right] = \hat{\sigma}_k^{11}\,e^{-\beta\epsilon_k(g,\gamma)} + \hat{\sigma}_k^{22}\,e^{\beta\epsilon_k(g,\gamma)} - e^{i\theta\mu_k} - e^{i\theta\lambda_k}. \quad (73)
$$

Using Proposition 2.5 we obtain

$$
2\tilde{P}^+(\theta) = \prod_{k\in\mathbf{k}^+}\left(\hat{\sigma}_k^{11}\,e^{-\beta\epsilon_k(g,\gamma)} + \hat{\sigma}_k^{22}\,e^{\beta\epsilon_k(g,\gamma)} + e^{i\theta\mu} + e^{i\theta\lambda}\right)
$$
$$
+ \prod_{k\in\mathbf{k}^+}\left(\hat{\sigma}_k^{11}\,e^{-\beta\epsilon_k(g,\gamma)} + \hat{\sigma}_k^{22}\,e^{\beta\epsilon_k(g,\gamma)} - e^{i\theta\mu} - e^{i\theta\lambda}\right). \quad (74)
$$

3. To determine $\tilde{P}^-(\theta)$ it remains to compute the contributions corresponding to $0, \pi$ momenta. Denoting

$$
\hat{w}_0 = \text{diag}\left(w_0^1, w_0^2\right), \quad \hat{w}_\pi = \text{diag}\left(w_\pi^1, w_\pi^2\right), \quad (75)
$$

one finds

$$
\hat{\rho}_0\exp\left(i\theta\hat{w}_0\right) = \text{diag}\left(e^{\beta(g-1)+i\theta w_0^1}, e^{-\beta(g-1)+i\theta w_0^2}\right),
$$
$$
\hat{\rho}_\pi\exp\left(i\theta\hat{w}_\pi\right) = \text{diag}\left(e^{\beta(g+1)+i\theta w_\pi^1}, e^{-\beta(g+1)+i\theta w_\pi^2}\right). \quad (76)
$$

Therefore, the negative-parity part of the characteristic function is

$$
2\tilde{P}^-(\theta) = \tilde{P}^F(\theta)\prod_{k\in\mathbf{k}^-}\left(\hat{\sigma}_k^{11}\,e^{-\beta\epsilon_k(g,\gamma)} + \hat{\sigma}_k^{22}\,e^{\beta\epsilon_k(g,\gamma)} + e^{i\theta\mu} + e^{i\theta\lambda}\right)
$$
$$
- \tilde{P}^B(\theta)\prod_{k\in\mathbf{k}^-}\left(\hat{\sigma}_k^{11}\,e^{-\beta\epsilon_k(g,\gamma)} + \hat{\sigma}_k^{22}\,e^{\beta\epsilon_k(g,\gamma)} - e^{i\theta\mu} - e^{i\theta\lambda}\right), \quad (77)
$$

where

$$
\tilde{P}^F(\theta) = \left(e^{\beta(g-1)+i\theta w_0^1} + e^{-\beta(g-1)+i\theta w_0^2}\right)\left(e^{\beta(g+1)+i\theta w_\pi^1} + e^{-\beta(g+1)+i\theta w_\pi^2}\right),
$$
$$
\tilde{P}^B(\theta) = \left(e^{\beta(g-1)+i\theta w_0^1} - e^{-\beta(g-1)+i\theta w_0^2}\right)\left(e^{\beta(g+1)+i\theta w_\pi^1} - e^{-\beta(g+1)+i\theta w_\pi^2}\right). \quad (78)
$$

Note that this is not the only way to calculate the characteristic function: instead of diagonalizing $\hat{\rho}_k$, one could diagonalize an observable $\hat{w}_k$. However, in our approach the role of the Boltzmann factor set by $\beta\epsilon_k(g,\gamma)$, which is usually dominant, is clear from the formulas (74) and (77). In the following sections we apply this method to characterize the full counting statistics of two physically important observables, the number of kinks and the transverse magnetization.

### 4.1 Probability distribution of the number of kinks at thermal equilibrium

We next derive the full generating function for the kink-number operator, which is of fundamental importance in the study of quantum phase transitions [12, 16, 27–30]. Although the relevance of this operator is most apparent in the Ising model, it is also well-defined for the general XY model. In the following, we consider the TFQIM with $\gamma = 1$ for simplicity. The explicit form of kink-number operator reads

$$\hat{N} = \frac{1}{2} \sum_{n=1}^{L} \left( 1 - \hat{X}_n \hat{X}_{n+1} \right), \tag{79}$$

with eigenvalues $n = 0, 1, \ldots, L$ under periodic boundary conditions.

Comparing the Ising Hamiltonian Eq. (1), with $\gamma = 1$ and $g = 0$, with the Bogoliubov Hamiltonian (18) at $\gamma = 1$ and $g = 0$, the kink operator takes a simple form as the sum of the number operators of quasiparticles in each momentum [12]. Here, we generalize the kink number operator definition for all values of the magnetic field. First, we rewrite the operator (79) in the following form:

$$\hat{N} = \frac{L}{2} + \sum_k \hat{N}_k. \tag{80}$$

By analogy with Eq. (65) and Eq. (66), we define a new set of operators $\hat{n}_k$, $\hat{n}_0$, and $\hat{n}_\pi$; taking for any mode $k \neq 0, \pi$ the basis given by $\{|00\rangle_k, |11\rangle_k, |01\rangle_k, |10\rangle_k\}$, while selecting for $0, \pi$ momenta the basis $\{|0\rangle_0, c_0^\dagger |0\rangle_0\}$, $\{|0\rangle_\pi, c_\pi^\dagger |0\rangle_\pi\}$. Therefore, we define the operators

$$\hat{n}_k = \begin{pmatrix} \cos(k) & \sin(k) & 0 & 0 \\ \sin(k) & -\cos(k) & 0 & 0 \\ 0 & 0 & 0 & 0 \\ 0 & 0 & 0 & 0 \end{pmatrix}, \quad \hat{n}_0 = \begin{pmatrix} \frac{1}{2} & 0 \\ 0 & -\frac{1}{2} \end{pmatrix} \quad \hat{n}_\pi = \begin{pmatrix} -\frac{1}{2} & 0 \\ 0 & \frac{1}{2} \end{pmatrix}, \tag{81}$$

and thus

$$\hat{n}_k^{(1)} = \begin{pmatrix} \cos(k) & \sin(k) \\ \sin(k) & -\cos(k) \end{pmatrix}, \quad \hat{n}_k^{(2)} = 0_2. \tag{82}$$

Note that $\exp\left( i\theta \hat{n}_k^{(1)} \right)$ has the simple form

$$\exp\left( i\theta \hat{n}_k^{(1)} \right) = \begin{pmatrix} \cos(\theta) + i\sin(\theta)\cos(k) & i\sin(\theta)\sin(k) \\ i\sin(\theta)\sin(k) & \cos(\theta) - i\sin(\theta)\cos(k) \end{pmatrix}. \tag{83}$$

Using expressions (68) and (72), one finds

$$\sigma_k^{11} = \cos(\theta) + i\sin(\theta)\cos(k - \vartheta_k),$$
$$\sigma_k^{22} = \cos(\theta) - i\sin(\theta)\cos(k - \vartheta_k). \tag{84}$$

This yields the explicit expression of the full characteristic function of the kink-number operator.

**Summary 4.1: Full characteristic function for kink number operator**

The full characteristic function of the kink number operator Eq. (79) at thermal equilibrium reads

$$\tilde{P}(\theta) = \frac{1}{Z(\beta,g,\gamma)}\left[\tilde{P}^+(\theta) + \tilde{P}^-(\theta)\right].$$  (85)

**Positive part of characteristic function:**

$$\tilde{P}^+(\theta) = \frac{\exp(iL\theta/2)}{2}$$

$$\times \left[ \prod_{k\in\mathbf{k}^+} 2\left(\cos(\theta)\cosh[\beta\epsilon_k(g,\gamma)] - i\sin(\theta)\sinh[\beta\epsilon_k(g,\gamma)]\cos(k-\vartheta_k) + 1\right)\right.$$

$$\left. + \prod_{k\in\mathbf{k}^+} 2\left(\cos(\theta)\cosh[\beta\epsilon_k(g,\gamma)] - i\sin(\theta)\sinh[\beta\epsilon_k(g,\gamma)]\cos(k-\vartheta_k) - 1\right)\right].$$  (86)

**Negative part of characteristic function:**

$$\tilde{P}^-(\theta) = \frac{\exp(iL\theta/2)}{2}$$

$$\times \left[ \tilde{P}^F(\theta)\prod_{k\in\mathbf{k}^-} 2\left(\cos(\theta)\cosh[\beta\epsilon_k(g,\gamma)] - i\sin(\theta)\sinh[\beta\epsilon_k(g,\gamma)]\cos(k-\vartheta_k) + 1\right)\right.$$

$$\left. - \tilde{P}^B(\theta)\prod_{k\in\mathbf{k}^-} 2\left(\cos(\theta)\cosh[\beta\epsilon_k(g,\gamma)] - i\sin(\theta)\sinh[\beta\epsilon_k(g,\gamma)]\cos(k-\vartheta_k) - 1\right)\right],$$  (87)

where

$$\tilde{P}^F(\theta) = 2^2\cosh\left(\frac{\beta\epsilon_{k=0} + i\theta}{2}\right)\cosh\left(\frac{\beta\epsilon_{k=\pi} - i\theta}{2}\right),$$

$$\tilde{P}^B(\theta) = 2^2\sinh\left(\frac{\beta\epsilon_{k=0} + i\theta}{2}\right)\sinh\left(\frac{\beta\epsilon_{k=\pi} - i\theta}{2}\right).$$  (88)

The exact total partition function is given by Eq. (48), with the eigenenergies $\epsilon_k(g,\gamma)$ and $\epsilon_{k=0}$ given by Eq. (47), and the Bogoliubov angles $\vartheta_k$ satisfying Eq. (17).

By contrast, in the customary PPA, the characteristic function of the kink-number operator in the thermodynamic limit contains only the first term of $\tilde{P}^+(\theta)$:

**Summary 4.2: PPA characteristic function for kink number**

In the thermodynamic limit, Eq. (85) reduces to

$$\tilde{P}_{\text{PPA}}(\theta) = \frac{\exp(iL\theta/2)}{Z_F^+(\beta,g,\gamma)}\prod_{k\in\mathbf{k}^+} 2\left(\cos(\theta)\cosh(\beta\epsilon_k(g,\gamma)) - i\sin(\theta)\sinh(\beta\epsilon_k(g,\gamma))\cos(k-\vartheta_k) + 1\right),$$  (89)

where $Z_F^+(\beta,g,\gamma)$ is defined in (49).

In Figure 3, we characterize the full counting statistics of kinks as a function of the magnetic field and inverse temperature. By numerical integration of Eq. (60), we find the exact probability distribution function $P(n)$ using Eq. (85). Additionally, we evaluate the PPA probability distribution function using Eq. (89). The use of the PPA partition functions is

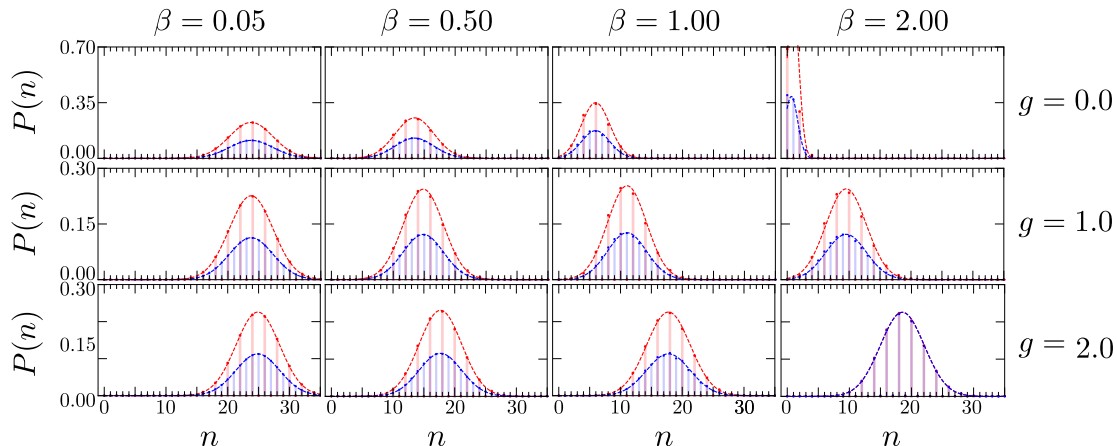

Figure 3: **Kink-number distribution at thermodynamic equilibrium.** Probability distribution of the number of kinks $P(n)$ as a function of the magnetic field $g$ and temperature $T$ for a chain of $L = 50$ spins. The exact probability distribution Eq. (85) (red bars) is compared with the simplified expression in Eq. (89) (blue bars). Only in the low-temperature paramagnet the PPA is accurate. Further, the normal (Gaussian) approximation to the histograms is also shown (dashed lines).

widely extended in the literature, e.g., to analyze the formation of kinks after non-equilibrium quenches [1, 44, 59, 60, 62, 64]. For a large magnetic field and low temperature, the PPA works well and reproduces essentially the exact full counting statistics of kinks. By contrast, when thermal fluctuations are suppressed and the magnetic field contribution dominates, the PPA leads to pronounced discrepancies (i.e. see Fig. 3 lower-left panels). The PPA also fails to account for momentum conservation. Under periodic boundary conditions, kinks appear in pairs. In general, the PPA incorrectly predicts a non-zero probability of exciting *odd* number of kinks:

$$P_{\text{PPA}}(n = 2\ell + 1) = \frac{1}{2\pi} \int_{-\pi}^{\pi} d\theta \tilde{P}(\theta) \exp\left[-i\theta(2\ell + 1)\right] \neq 0, \qquad (90)$$

but for large $g$ and $\beta$ as shown in 3, when $P_{\text{PPA}}(n = 2\ell + 1) \approx 0$.

The fact that only even number of kinks in the presence of periodic boundary conditions can be excited is intuitively clear. For a simple mathematical argument, consider the operator $\prod_{n=1}^{L} \hat{X}_n \hat{X}_{n+1}$ which is 1 for even kink number and $-1$ for an odd number. Using $\hat{X}_{L+1} = \hat{X}_1$ and $(\hat{X}_n)^2 = \bigotimes_{n=1}^{L} \hat{\mathbb{I}}_n$, it satisfies:

$$\prod_{n=1}^{L} \hat{X}_n \hat{X}_{n+1} = 1. \qquad (91)$$

The PPA characteristic function, $\tilde{P}^+(\theta)$ and $\tilde{P}^-(\theta)$ do not exhibit this feature.

In addition, we note that the magnitude of the exact $P(n)$ for even $n$ can be approximated by the coarse-grained PPA approximation, whenever the distribution is symmetric, with tails far from the origin, i.e.,

$$P(n) \approx P_{\text{PPA}}(n) + \frac{1}{2}\left[P_{\text{PPA}}(n-1) + P_{\text{PPA}}(n+1)\right], \qquad (92)$$

as shown in Figure 4.

An analysis of the cumulants of the kink-number distribution as a function of the inverse temperature is presented in Fig. 5 for various system sizes. In the paramagnetic phase ($g > 1$),

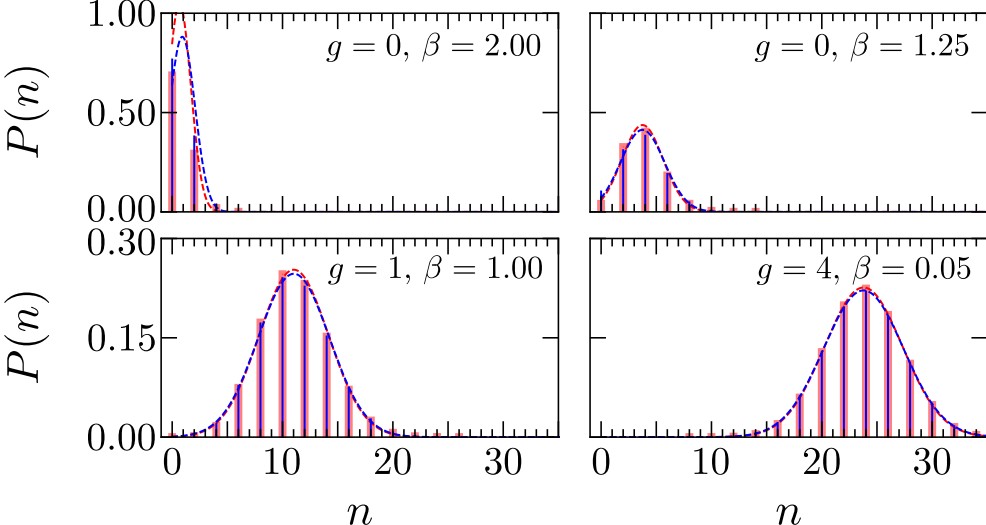

Figure 4: **Exact and Coarse-grained PPA kink-number probability distributions at thermal equilibrium.** The exact kink-number probability distribution evaluated using Eq. (85) (red) is compared with the coarse-grained PPA probability distribution Eq. (92) (blue). The numerical histograms are compared with the Gaussian $N(\kappa_1, \kappa_2)$ with fitted numerical values for $\kappa_1$ and $\kappa_2$ (dashed lines). In as much as the exact distribution is symmetric and its left tail is negligible near the origin, the coarse-graining of the PPA distribution in Eq. (92) reproduces accurately the exact distribution. Deviations are manifested at low $g$ and temperature, when the distribution is asymmetric.

the mean always exceeds the variance, making the kink-number distribution sub-Poissonian. This need not be the case in the ferromagnetic phase, where the distribution changes from sub-Poissonian to super-Poissonian as the temperature decreases. This behavior is shown to be robust as a function of the system size. The difference between the exact cumulant values and those derived from the PPA is systematically studied in Fig. 6 for a system size of $L = 12$ spins; the relative error is reduced with increasing system size. The quality of the PPA improves with increasing temperature, in the classical regime, in the ferromagnetic phase. While the dependence of the relative error as a function of the magnetic field $g$ is not monotonic, the bigger discrepancies between the exact results and the PPA are found in the ferromagnetic phase in the low temperature regime, when the relative error can reach 100%. In the paramagnetic phase, the PPA provides an accurate description of the cumulants for different temperatures and values of the magnetic field

To complete the characterization of the kink-number distribution we consider the limiting cases of the ground-state distribution ($\beta \to \infty$) and the infinite-temperature case ($\beta \to 0$) in an exact approach, without using the PPA. The first can be easily described using (83), while in the second we consider a maximally-mixed Gibbs state and apply trace formulas 2.5. For $\beta = 0$, the exact result and the PPA coincide.

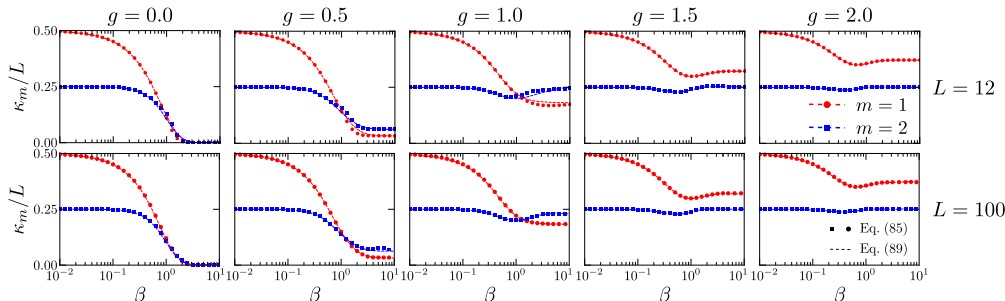

**Figure 5: Cumulants of the kink-number distribution as a function of the inverse of temperature $\beta$.** Using the exact characteristic function given by Eq. (85), the mean kink number $\kappa_1$ and the variance $\kappa_2$ are shown by red circles and blue squares, respectively. The dashed lines correspond to the numerical results using the PPA characteristic function in Eq. (89). While in the paramagnetic phase the statistics is sub-Poissonian, in the ferromagnetic phase it changes from sub- to super-Poissonian as the temperature is decreased. The magnetic field is increased from 0.0 to 2.0, varying from left to right in steps of 0.5. In the upper panels, the system size is $L = 12$, while in the lower ones $L = 100$.

---

**Summary 4.3: Limiting cases of kink number distribution**

**Exact ground-state characteristic function of the kink-number distribution:**

$$\tilde{P}_{\beta\to\infty}(\theta) = \exp(iL\theta/2) \prod_{k\in\mathbf{k}^+} (\cos\theta - i\sin\theta\cos(k-\vartheta_k)). \tag{93}$$

**Exact infinite-temperature characteristic function of the kink-number distribution:**

$$\tilde{P}_{\beta\to 0}(\theta) = \exp(iL\theta/2) \left( \cos^L\frac{\theta}{2} + (-1)^{L/2}\sin^L\frac{\theta}{2} \right). \tag{94}$$

---

Instances of the corresponding distributions are shown in Fig. 7 for the (pure) ground-state as a function the magnetic field. For $g = 0$ one finds a Kronecker delta distribution centered at $n = 0$, with $P(0) = 1$ and $P(n) = 0$ for $n > 1$, as expected. As the magnetic field is cranked up, the distribution broadens and gradually shifts away from the origin, becoming approximately symmetric in the paramagnetic phase.

The right panel in Fig. 7 also shows the corresponding distribution in the infinite-temperature case, that is symmetric, centered at $n = L/2$ and independent of the transverse magnetic field $g$, as can be seen from Eq. (94). In fact, full probability distribution for infinite temperature can be found by a combinatorial argument. Working in the basis of eigenstates of $\sigma_i^x$ in each site, the probability of obtaining $n = 2l$ kinks is related to the number of basis vectors with $2l$ spin flips, where we use the fact that an even number of kinks is enforced by boundary conditions. One can choose the location of $2l$ kinks in the chain in $2\binom{L}{2l}$ ways. Therefore, the full probability distribution has the form:

$$P_{\beta\to 0}(n = 2l) = \frac{1}{2^{L-1}}\binom{L}{2l}, \quad l = 0, 1, \dots \frac{L}{2}. \tag{95}$$

The corresponding cumulant values read

$$\kappa_1 = \frac{L}{2}, \quad \kappa_2 = \frac{L}{4}, \quad \kappa_3 = 0, \quad \kappa_4 = -\frac{L}{8}, \quad \kappa_5 = 0, \quad \kappa_6 = \frac{L}{4}, \quad \dots \tag{96}$$

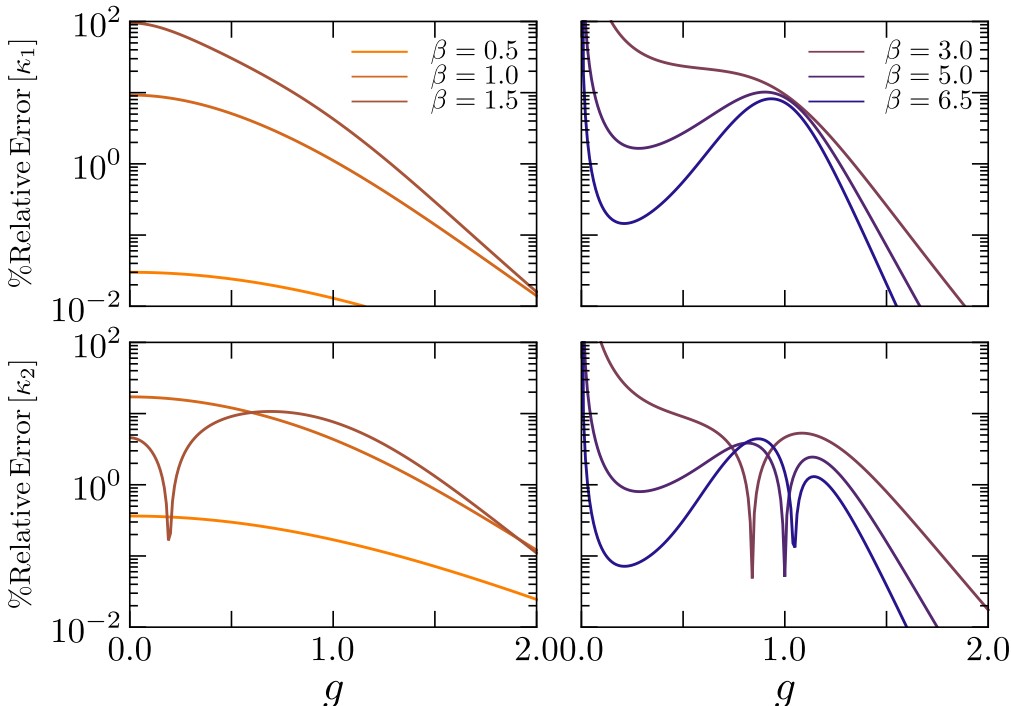

Figure 6: **Relative error for the first two cumulants of the kink-number distribution as a function of magnetic field** $g$. Using the full characteristic function in Eq. (85) and the PPA characteristic function Eq. (89), the relative error is evaluated as a function of the magnetic field for a system size $L = 12$ and different temperatures.

By keeping the first two cumulants and setting the rest to zero, $P_{\beta \to 0}(n = 2l)$ can be approximated by a Gaussian distribution $N(\kappa_1, \kappa_2)$ with mean $\kappa_1 = L/2$ and variance $\kappa_2 = L/4$. As shown in Fig. 7 this approximation describes the envelope of the distribution with great accuracy.

## 4.2 Probability distribution for the transverse magnetization at thermal equilibrium

We next focus on the derivation of the explicit form of the characteristic function of the transverse magnetization

$$\hat{M} = \sum_{n=1}^{L} \hat{Z}_n \,, \tag{97}$$

with eigenvalues $m = -L, -L + 2, \dots, L - 2, L$ for even $L$. The latter has been studied in the PPA and continuous approximations and finds broad applications in the characterization of quantum critical behavior [82–86, 88, 89] and the identification of various many-body states in ultracold-atom quantum simulators [87].

In the Fourier representation, it is the sum of two different contributions:

$$\hat{M}^+ = \sum_{k \in \mathbf{k}^+} 2(\hat{c}_k \hat{c}_k^\dagger - \hat{c}_k^\dagger \hat{c}_k), \quad \hat{M}^- = \sum_{k \in \mathbf{k}^-} 2(\hat{c}_k \hat{c}_k^\dagger - \hat{c}_k^\dagger \hat{c}_k) + \hat{c}_0 \hat{c}_0^\dagger - \hat{c}_0^\dagger \hat{c}_0 + \hat{c}_\pi \hat{c}_\pi^\dagger - \hat{c}_\pi^\dagger \hat{c}_\pi \,. \tag{98}$$

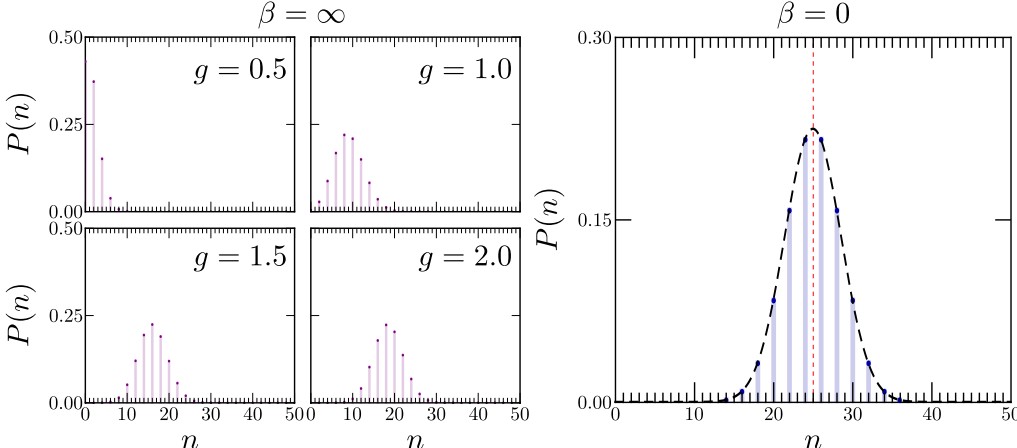

Figure 7: **Limiting cases of kink number distribution.** Probability distribution of the number of kinks $P(n)$ as a function of the magnetic field $g$ and inverse temperature $\beta$ for a chain of $L = 50$ spins. The left panel shows the kink-number distribution for different values of the magnetic field and is obtained using the ground-state characteristic function Eq. (93). The right panel shows the kink number distribution at infinity temperature, computed using the characterization function given by Eq. (94). The vertical dashed red line is located at $\kappa_1 = L/2$, while the long-dashed black line corresponds to the Gaussian approximation $N(L/2, L/4)$.

In parallel with Eq. (81), we define a new set of a single-mode operators $\hat{m}_k$, $\hat{m}_0$, and $\hat{m}_\pi$,

$$
\hat{m}_k = \begin{pmatrix} 2 & 0 & 0 & 0 \\ 0 & -2 & 0 & 0 \\ 0 & 0 & 0 & 0 \\ 0 & 0 & 0 & 0 \end{pmatrix}, \qquad \hat{m}_k^{(1)} = \begin{pmatrix} 2 & 0 \\ 0 & -2 \end{pmatrix}, \qquad \hat{m}_k^{(2)} = 0_2. \tag{99}
$$

In addition, in the negative-parity sector, the matrix $\hat{m}_k$ has the same form for the momenta $0, \pi$ that is given by $\hat{m}_0 = \hat{m}_\pi = \text{diag}(1, -1)$. We can easily compute $\exp\left(i\theta \hat{m}_k^{(1)}\right)$ and the $\hat{\sigma}_k$ matrix to obtain

$$
\begin{aligned}
\hat{\sigma}_k^{(11)} &= \cos(2\theta) + i\cos(\vartheta_k)\sin(2\theta), \\
\hat{\sigma}_k^{(22)} &= \cos(2\theta) - i\cos(\vartheta_k)\sin(2\theta).
\end{aligned} \tag{100}
$$

**Summary 4.4: Full generating function of transverse magnetization**

The full characteristic function for the transverse magnetization Eq. (97) at thermal equilibrium reads

$$\tilde{P}(\theta) = \frac{1}{Z(\beta,g,\gamma)} \left( \tilde{P}^+(\theta) + \tilde{P}^-(\theta) \right). \tag{101}$$

**Positive part of characteristic function:**

$$\tilde{P}^+(\theta) = \frac{1}{2}\Bigg[ \prod_{k\in\mathbf{k}^+} 2\left(\cos(2\theta)\cosh(\beta\epsilon_k(g,\gamma)) - i\sin(2\theta)\sinh(\beta\epsilon_k(g,\gamma))\cos(\vartheta_k) + 1\right)$$

$$+ \prod_{k\in\mathbf{k}^+} 2\left(\cos(2\theta)\cosh(\beta\epsilon_k(g,\gamma)) - i\sin(2\theta)\sinh(\beta\epsilon_k(g,\gamma))\cos(\vartheta_k) - 1\right) \Bigg]. \tag{102}$$

**Negative part of characteristic function:**

$$\tilde{P}^-(\theta) = \frac{1}{2}\Bigg[ \tilde{P}^F(\theta) \prod_{k\in\mathbf{k}^-} 2\left(\cos(2\theta)\cosh(\beta\epsilon_k(g,\gamma)) - i\sin(2\theta)\sinh(\beta\epsilon_k(g,\gamma))\cos(\vartheta_k) + 1\right)$$

$$- \tilde{P}^B(\theta) \prod_{k\in\mathbf{k}^-} 2\left(\cos(2\theta)\cosh(\beta\epsilon_k(g,\gamma)) - i\sin(2\theta)\sinh(\beta\epsilon_k(g,\gamma))\cos(\vartheta_k) - 1\right) \Bigg], \tag{103}$$

with

$$\tilde{P}^F(\theta) = 2^2 \cosh\left(\frac{\beta\epsilon_{k=0} + 2i\theta}{2}\right)\cosh\left(\frac{\beta\epsilon_{k=\pi} + 2i\theta}{2}\right),$$

$$\tilde{P}^B(\theta) = 2^2 \sinh\left(\frac{\beta\epsilon_{k=0} + 2i\theta}{2}\right)\sinh\left(\frac{\beta\epsilon_{k=\pi} + 2i\theta}{2}\right). \tag{104}$$

The exact partition function is given by Eq. (48), with the eigenenergies $\epsilon_k(g,\gamma)$ and $\epsilon_{k=0}$ given by Eq. (47), and the Bogoliubov angles $\vartheta_k$ satisfying Eq. (17).

By contrast, in the PPA, the characteristic function of the transverse magnetization in the thermodynamic limit contains only the first term of $\tilde{P}^+(\theta)$:

**Summary 4.5: PPA characteristic function for transverse magnetization**

In the thermodynamic limit, Eq. (101) reduces to

$$\tilde{P}_{\text{PPA}}(\theta) = \frac{1}{2Z_F^+(\beta,g,\gamma)} \prod_{k\in\mathbf{k}^+} 2\left(\cos(2\theta)\cosh(\beta\epsilon_k(g,\gamma)) - i\sin(2\theta)\sinh(\beta\epsilon_k(g,\gamma))\cos(\vartheta_k) + 1\right), \tag{105}$$

where $Z_F^+(\beta,g,\gamma)$ is defined in (49).

The magnetization distribution is shown in Fig. 8 for different values of $g$ and $\beta$ for a fixed system size $L = 50$. The distribution $P(m)$ vanishes for odd values of $m$ for even $L$. It is naturally symmetric for $g = 0$ and approximately so for finite $g$ in the high-temperature case at low magnetic fields, when it approaches a binomial distribution. The accuracy of the PPA is remarkable as a function of $g$ and $\beta$ with discrepancies being noticeable in the pure ferromagnet ($g = 0$) at low temperature. As the magnetic field is cranked up at constant $\beta$, the alignment of the spins is favored shifting the mean and increasing the negative skewness of the distribution in the paramagnetic phase.

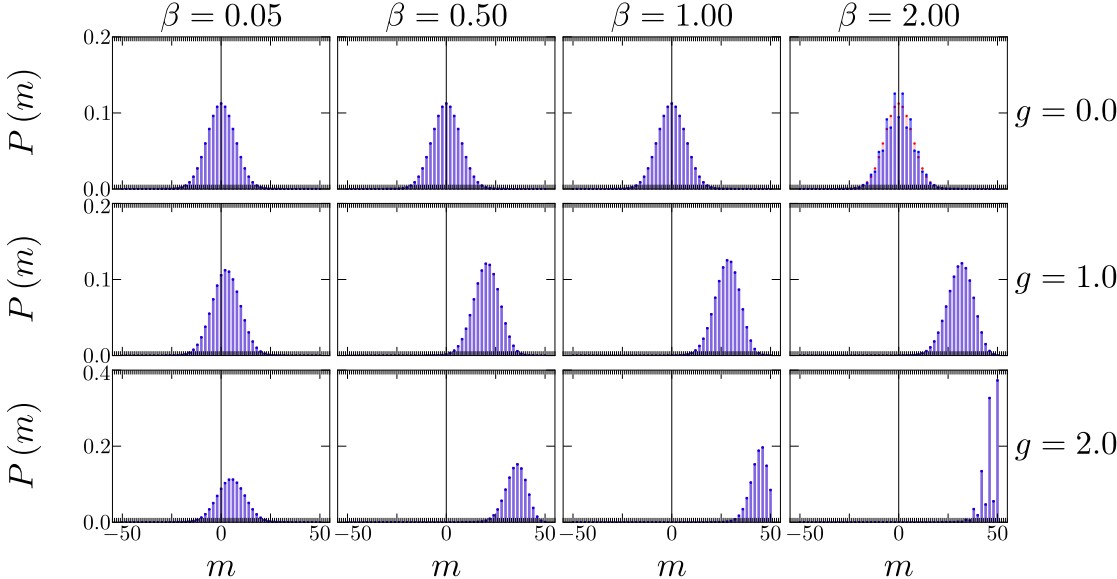

Figure 8: **Magnetization distribution at thermodynamic equilibrium.** Probability distribution of the transverse magnetization $P(m)$ for different values of the magnetic field $g$ and inverse temperature $\beta$ in a chain of $L = 50$ spins. The exact probability distribution Eq. (101) (red bars) is compared with the simplified expression in Eq. (105) (blue bars).

Figure 9 provides a systematic characterization of the first two cumulants as a function of the inverse temperature for different values of $g$. In contrast with the kink-number distribution, in the ferromagnetic phase the variance always exceeds the mean, and thus the magnetization distribution remains super-Poissonian. In the paramagnetic phase, at any fixed value of $g$ the variance decreases with temperature, while the converse is true for the mean magnetization. As a result, the character of the distribution changes from super-Poissonian to sub-Poissonian as the the temperature is lowered. The behavior of $P(m)$ is shown to be robust as a function of the system size, with discrepancies between the exact results and the PPA being restricted to the critical point. The relative error of the PPA remains below 10% as a function of $g$ and $\beta$ as shown in Fig. 10.

As in the case of kink number distribution, we close with a characterization of the magnetization distribution in the limits of infinite and vanishing inverse temperature $\beta$.

---

**Summary 4.6: Limiting cases of transverse magnetization distribution**

**Exact ground-state characteristic function of transverse magnetization**:

$$\tilde{P}_{\beta\to\infty}(\theta) = \prod_{k\in\mathbf{k}^+}(\cos 2\theta - i\sin 2\theta \cos\vartheta_k).\qquad(106)$$

**Exact infinite-temperature characteristic function of transverse magnetization**:

$$\tilde{P}_{\beta\to 0}(\theta) = \cos^L\theta.\qquad(107)$$

---

The behavior of the ground-state magnetization distribution is the reverse of the kink-number distribution in the sense that it becomes approximately symmetric in the ferromagnetic phase and sharply peaked at $m = L$ in the paramagnetic phase. Using formulas (106) and (63),

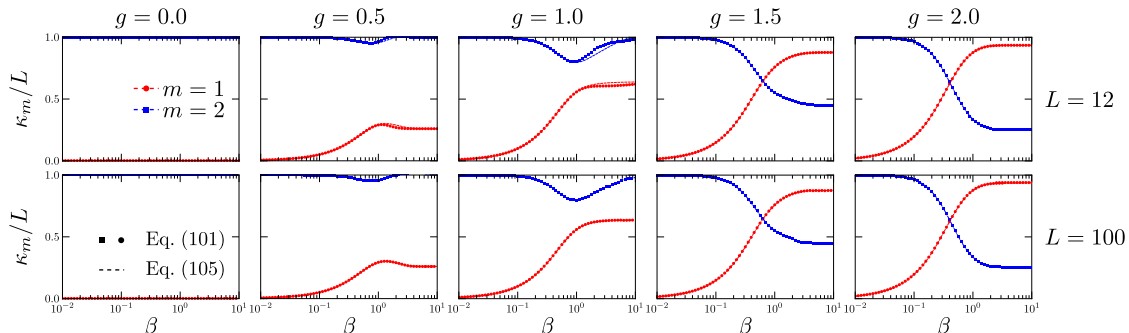

Figure 9: **Cumulants of the magnetization distribution as a function of the inverse of temperature $\beta$.** Using the full characteristic function given by Eq. (101), the mean value of the transversal magnetization $\kappa_1$ and the variance $\kappa_2$ are shown by red circles and blue squares, respectively. The dashed lines correspond to the numerical results using the simplified characteristic function (Eq. (105)). In the ferromagnetic phase the statistics is super-Poissonian, while it changes from super- to sub-Poissonian in the paramagnetic phase as the temperature is decreased. The magnetic field varies from 0.0 to 2.0 from left to right in steps of 0.5. The system size is $L = 12$ in the upper row and $L = 100$ in the lower one.

one can find the first cumulants of the ground-state distribution explicitly. In particular, the first few cumulants read

$$\kappa_1 = -\sum_{k \in \mathbf{k}^+} 2 \cos \vartheta_k , \tag{108}$$

$$\kappa_2 = L - 2 \sum_{k \in \mathbf{k}^+} \cos(2\vartheta_k) , \tag{109}$$

$$\kappa_3 = 4 \sum_{k \in \mathbf{k}^+} [\cos(\vartheta_k) - \cos(3\vartheta_k)] . \tag{110}$$

The second cumulant turns out to have a particularly simple form due to its close relation to the ground-state fidelity susceptibility [74, 94] and reads

$$\kappa_2 = L \frac{1 + g^{L-2}}{1 + g^L} . \tag{111}$$

By contrast, in the infinite-temperature case, in which the PPA is exact, the distribution is symmetric, centered at $m = 0$ and independent of the magnetic field. The magnetization distribution describes in this case the sum of $L$ independent discrete random variables with outcomes $\pm 1$ with equal probability $1/2$. As a result $\kappa_1 = 0$, $\kappa_2 = L/4$. In the infinite temperature limit, $P(m)$ is equal to that of a classical Ising chain and can be written explicitly:

$$P_{\beta \to 0}(m) = \frac{1}{2^L} \binom{L}{\frac{1}{2}(m+L)}, \quad m = -L, -L+2, \ldots, L-2, L . \tag{112}$$

Odd cumulant identically vanish, while the first even ones read

$$\kappa_2 = L , \quad \kappa_4 = -2L , \quad \kappa_6 = 16L , \quad \kappa_8 = -272L , \quad \kappa_{10} = 7936L , \quad \ldots \tag{113}$$

As a result, in the normal approximation $P_{\beta \to 0}(n = 2l)$ is given by Gaussian distribution with zero mean and variance $\kappa_2 = L$. Fig. 11 shows this Gaussian distribution as a black envelope, accurately approximating the exact results.

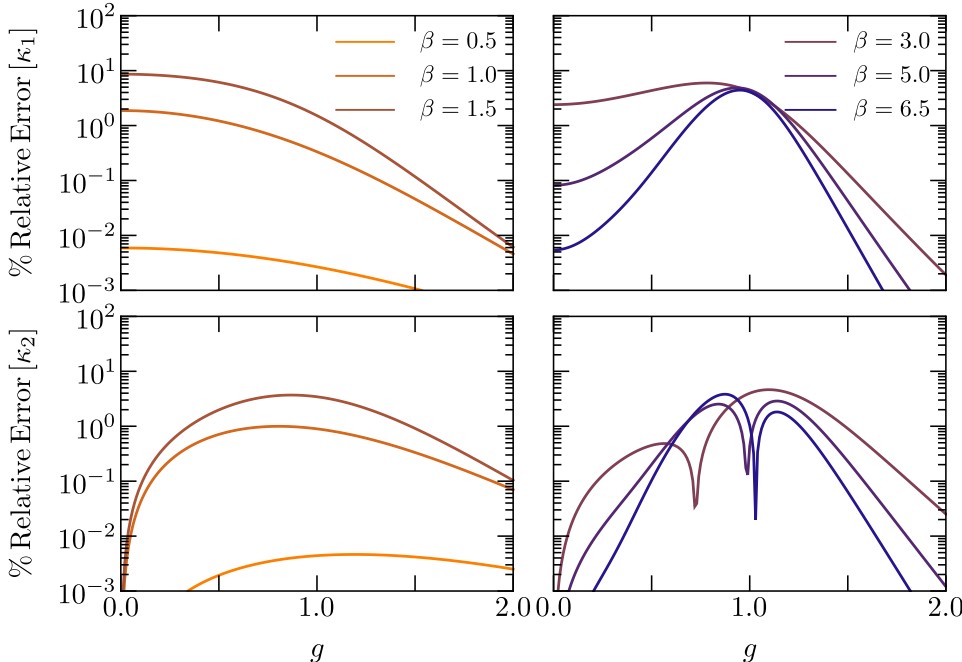

Figure 10: **Relative error for the first two cumulants of the magnetization distribution as a function of magnetic field** $g$. Using the full characteristic function in Eq. (101) and the corresponding PPA Eq. (105), the relative error is evaluated as a function of the magnetic field for a system size $L = 12$ and different temperatures.

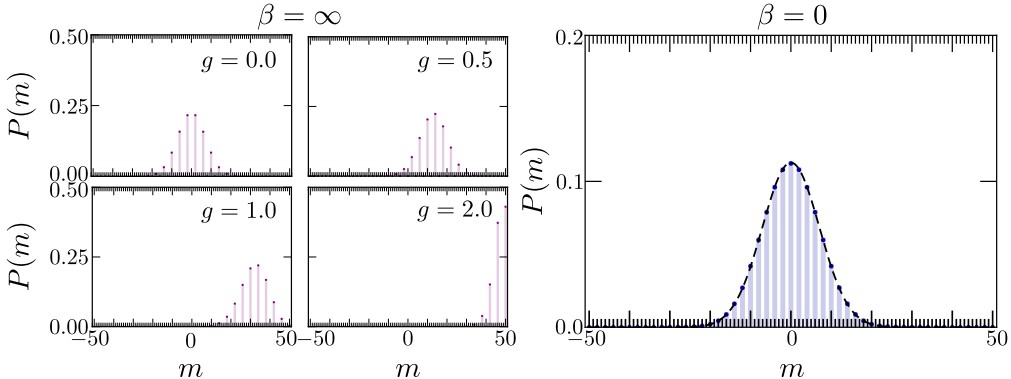

Figure 11: **Limiting cases of transverse magnetization distribution.** Probability distribution of the transverse magnetization $P(m)$ as a function of the magnetic field $g$ and inverse temperature $\beta$ for a chain of $L = 50$ spins. The left panel shows the ground-state transverse magnetization distribution for different values of the magnetic field, and is computed using the characteristic function Eq. (106). The right panel shows the transverse magnetization distribution at infinity temperature, obtained using the characterization function given by Eq. (107). The envelope of the distribution is reproduced by the Gaussian approximation $N(0, L)$ shown as a dashed black line.

## 5 Conclusion

We have provided an exact treatment of the thermal equilibrium properties for a class of integrable spin chains that admit a description in terms of free fermions. Instances of this family are the one-dimensional transverse-field Ising, XY and Kitaev models, among other examples. Whenever the system Hamiltonian commutes with parity operator, the complete Hilbert spaces is the direct sum of the corresponding even and odd parity subspaces. For an exact treatment of thermal equilibrium, we have detailed an algebraic approach in the complete Hilbert spaces and provided the exact expression for the partition function. We have identified the limitations of the approximate description of thermal equilibrium in terms of the positive-parity sector, frequently adopted in the literature. This approximate approach fails in what can be considered the most interesting regime: the neighborhood of a quantum critical point at low temperatures. In particular, we have shown that the discrepancies between the exact and approximate results can lead to significant errors in this regime.

Making use of the exact algebraic framework, we have computed as well the eigenvalue probability distribution of different observables. As an application, we have characterized in detail the distribution of the number of kinks as well as the transverse magnetization, covering all regimes from zero temperature (ground-state behavior) to infinite temperature. Our results are of direct relevance to the study of thermal equilibrium properties of integrable spin chains as well as the study of the nonequilibrium dynamics generated by driving a thermal state out of equilibrium. They are thus expected to find applications in the description of quantum simulation experiments, quantum annealing and quantum thermodynamics of spin systems. As a prospect, it is interesting to extend our results to the generalized Gibbs state whenever the relaxing dynamics of an initial state preserves a set of integrals of motion.

## Acknowledgements

We thank Bogdan Damski, Jacek Dziarmaga and Victor Mukherjee for insightful remarks concerning the manuscript.

**Funding information** We acknowledge funding support from the Spanish Ministerio de Ciencia e Innovación (PID2019-109007GA-I00). MB acknowledges support of Polish National Science Center (NCN) grant DEC-2016/23/B/ST3/01152, NCN scholarship ETIUDA 8 (2020/36/T/ST3/0032) and Faculty of Physics, astronomy and Computer Science internal grant N17/MNS/000008.

## A   Proof Proposition 2: Identities for Traces

First, the formulas given by Eq. (34) are true for $n = 1$. We assume that they are true for some $n \geq 1$ and we compute

$$\text{tr}\left[\mathcal{P}\left(\bigotimes_{i=1}^{n+1}\hat{O}_{k_i}\right)\right] = \text{tr}\left[\mathcal{P}\left(\bigotimes_{i=1}^{n}\hat{O}_{k_i}\right)\right]\cdot\text{tr}\left[\hat{O}_{k_{n+1}}^{(p)}\right] + \text{tr}\left[\mathcal{N}\left(\bigotimes_{i=1}^{n}O_{k_i}\right)\right]\cdot\text{tr}\left[\hat{O}_{k_{n+1}}^{(n)}\right]$$

$$= \frac{1}{2}\left[\left(\text{tr}\left[\hat{O}_{k_{n+1}}^{(p)}\right] + \text{tr}\left[\hat{O}_{k_{n+1}}^{(n)}\right]\right)\prod_{i=1}^{n}\text{tr}\left[\hat{O}_{k_i}\right]\right.$$

$$\left. + \left(\text{tr}\left[\hat{O}_{k_{n+1}}^{(p)}\right] - \text{tr}\left[\hat{O}_{k_{n+1}}^{(n)}\right]\right)\prod_{i=1}^{n}\left(\text{tr}\left[\hat{O}_{k_i}^{(p)}\right] - \text{tr}\left[\hat{O}_{k_i}^{(n)}\right]\right)\right]$$

$$= \frac{1}{2}\left[\prod_{i=1}^{n+1}\text{tr}\left[\hat{O}_{k_i}\right] + \prod_{i=1}^{n+1}\left(\text{tr}\left[\hat{O}_{k_i}^{(p)}\right] - \text{tr}\left[\hat{O}_{k_i}^{(n)}\right]\right)\right],$$

and an inductive step is completed.

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
