# Peer review of "Exact Thermal Properties of free-fermionic Spin Chains"

_SciPost Physics, doi:SciPost Phys. 11, 013 (2021)_

## Round 1 · Referee Report · Anonymous (Referee 1) · 2021-4-19

Report

1) Overall, it is certainly an illuminating exercise to be able to work our thermal states of quadratic fermionic systems but these thermal states are not as interesting as they might seem to be because these integrable systems do not thermalize. At best they can dephase towards a generalized Gibbs ensemble which should not be mistaken for the traditional Gibbs state considered in this work.

2) It seems that a ``positive parity approximation'' is being criticised both in the abstract and the introduction. However, the parity is a good quantum number so if the system is initialized in, say, a positive parity state then it will remain in the positive parity subspace during its evolution and it is no approximation at all. Therefore, one should be very carefull not to criticise works where it is not an approximation but a simplification thanks to the conservation law.

The main added value of the manuscript is the formalism that allows to work out the thermal distributions which is a non-trivial task. For this reason it is worth publication in SciPost after the authors carefully rewrite the introduction in order to avoid misleading comments.

  • validity: good
  • significance: good
  • originality: good
  • clarity: top
  • formatting: perfect
  • grammar: excellent

Author:  Michał Białończyk  on 2021-05-21  [id 1448]

(in reply to Report 1 on 2021-04-19)
Category:
reply to objection

,,Overall, it is certainly an illuminating exercise to be able to work our thermal states of quadratic fermionic systems but these thermal states are not as interesting as they might seem to be because these integrable systems do not thermalize. At best they can dephase towards a generalized Gibbs ensemble which should not be mistaken for the traditional Gibbs state considered in this work.''

Reply: We thank the referee for emphasizing the relevance of our study.
We agree with the referee that the generalized Gibbs state is the relevant ensemble in many scenarios. This is the case when the time-evolution preserves the conserved quantities so that these are indeed invariants of motion. In fermionic models, this is certainly the case under unitary evolution, in the absence of coupling to a surrounding environment.
However, the canonical Gibbs state we consider is the equilibrium state that results from the coupling of the Ising chain to a thermal environment (in weak-coupling) when the dynamics is not unitary and does not preserve the integrals of motion. Such thermal state has manifold applications indicated in the manuscript. Quantities that are conserved under unitary dynamics (fermionic occupation numbers, that become integrals of motion in this scenario) are not conserved under generic open quantum dynamics (nonunitary evolution), in the presence of a surrounding environment.
In short, the relevant ensemble can be the generalized or the standard Gibbs ensemble, depending on whether the thermalizing time-evolution preserves integrals of motion other than the energy or not.
We have added the following sentence to the end of the Discussion section:
“As a prospect, it is interesting to extend our results to the generalized Gibbs state whenever the relaxing dynamics of an initial state preserves a set of integrals of motion.”

,,It seems that a positive parity approximation is being criticised both in the abstract and the introduction. However, the parity is a good quantum number so if the system is initialized in, say, a positive parity state then it will remain in the positive parity subspace during its evolution and it is no approximation at all. Therefore, one should be very carefull not to criticise works where it is not an approximation but a simplification thanks to the conservation law.''

Reply: We have been careful in this regard. All the cited references (“[1–3, 43, 58–62]”) used approximate results and either consider the partition function using the PPA [1-3] or canonical thermal states using the PPA or both [43]. The exact treatment in all these cases involves the consideration of both subspaces, as no particular simplification occurs in the considered scenarios. We have taken the chance to add other seminal works that use the PPA in the thermodynamic limit and emphasized that not even then the PPA is exact. We have emphasized classic references such as the work by Pfeuty and the textbooks by Sachdev and Suzuki et al. to give a better account of the widespread use of this approximation.

,,The main added value of the manuscript is the formalism that allows to work out the thermal distributions which is a non-trivial task. For this reason it is worth publication in SciPost after the authors carefully rewrite the introduction in order to avoid misleading comments.''

Reply: We thank the referee for the positive assessment of our work.

Anonymous on 2021-05-22  [id 1456]

(in reply to Michał Białończyk on 2021-05-21 [id 1448])

As I already wrote in my first report, this is a piece of solid work that definitely is worth publication. My only critical comments concerned introduction/motivation. Now that they have been satisfactorily fixed I have no longer any objections. I am impressed by the in depth response to Referee 2. I would suggest publication without necessity for any further ammendmends.

---

## Round 1 · Referee Report · Ning Wu (Referee 2) · 2021-5-11

Weaknesses

The expressions for the obtained exact partition function seem already exist in previous literature.

Report

The authors focus on thermal properties of one-dimensional quantum spin chains that admit a free-fermion representation via the Jordan-Wigner transformation. Explicitly, they consider a finite-size quantum XY spin chain under periodic boundary conditions. The resulting even and odd fermion parity sectors are carefully treated to obtain explicit expression of the exact partition function, which is compared with the one in the even parity sector only for different temperatures and transverse fields. The discrepancy between the two in the low-temperature regime is explained by a two-level approximation. In addition, the full counting statistic of observables that preserves the fermion parity (such as the transverse magnetization and the number of kinks) is provided.

Investigation of finite-temperature and non-equilibrium properties of finite-size integrable spin chains in a mathematically rigorous way is important to the understanding of various concepts in statistical mechanics and mathematical physics. The manuscript is clearly written and the results are reliable. I believe the paper is worth being published in SciPost Physics, though have several comments the authors may wish to address.

  1. The derived exact partition function for the spin-1/2 XY chain given by Eq. (46) seems quite similar to Eq. (1.46) in Minoru Takahashi's book [M. Takahashi, Thermodynamics of one-dimensional solvable models (Cambridge University Press, Cambridge, 1999)]. The authors my wish to point out the connection/difference between the two results.

  2. On page 15 it is mentioned that the treatment of observables having components linear in fermion operators is beyond the scope of the present paper. The authors should comment further on the possible difficulties in obtaining the full counting statistics of such kind of operators.

  • validity: good
  • significance: good
  • originality: ok
  • clarity: high
  • formatting: perfect
  • grammar: excellent

Author:  Michał Białończyk  on 2021-05-21  [id 1447]

(in reply to Report 2 by Ning Wu on 2021-05-11)
Category:
answer to question

Due to the significant number of accompanying graphs, we submit the full response in one .pdf file.

Attachment:

scipost_202104_00014_v1.pdf

---

## Round 2 · Referee Report · Ning Wu (Referee 2) · 2021-5-25

Report

In their reply letter, the authors made a comprehensive comparison between their results and Takahashi's. They also made a more appropriate reference to S. Katsura's early work in the revised manuscript. I therefore recommend publication of the paper in SciPost Physics.

---

## Round 2 · Author Response

Dear Editor,

We have taken into account remarks of the referees and changed the manuscript accordingly (precise responses are in the answers to reports section). We hope that it will better fit to publication in SciPost now.

Yours sincerely,
Michał Białończyk, Fernando Gómez-Ruiz, Adolfo del Campo

---

## Round 2 · List of Changes

- at the end of Discussion section, we added: “As a prospect, it is interesting to extend our results to the generalized Gibbs state whenever the relaxing dynamics of an initial state preserves a set of integrals of motion.”

- We corrected the mistake in equation (5) - it should be "L" instead of "L+1" in second line. Moreover, there were two daggers missing, we corrected it.

- We expanded the paragraph after equation (66) according to second referee report

- We added references, mainly the reference to Takahashi's book (according to the remark of referee).

---

## Round 3 · Author Response

Dear SciPost Editorial College,
We address here the comments in the Editorial Report:

Editors write: “The Scipost editorial college unanimously agreed that your manuscript
deserved to be published on Scipost. However, the college thinks that the title (and bit also
the emphasis) is misleading about the content of the paper. A paper with title "Exact
Thermal Properties of Integrable Spin Chains" cannot be based only on the analysis of a free
fermionic chain. The college asks then to reconsider the title of the paper into something like
"Exact Thermal Properties of free-fermionic Spin Chains", stressing that it would be very
interesting to understand in more generality which of properties discussed here survive to
the addition of (integrable) interactions.”

Authors’ Reply: Following the suggestion by the editors, we hereby resubmit the
manuscript with the modified title "Exact Thermal Properties of free-fermionic Spin Chains",

Editors write: “Furthermore, some members of the college noticed that while the
bibliography is rather complete on the free fermion side, there is not a single reference to the
large literature on full-counting statistics in truly interacting integrable models (like XXZ spinchains
and 1D Bose gas). The paper will surely benefit of the inclusion of a few references on
the subject .”

Authors’ Reply: We appreciate the suggestion from some college members to include new
references about XXZ spin-chains and 1D Bose gas. However, since our manuscript describes
the thermal properties of the anisotropic XY spin chain in the Ising universal class, it would
be self-contradictory to include references from broader topic while restricting the title.
Therefore, we consider the bibliography complete and covering all relevant works in such
systems, beyond the scope of the manuscript.

Finally, we would appreciate if the editors could handle the manuscript promptly to prevent
further delays. Please note between the reception of final referee comments and editorial
decision a period of 34 days passed by in which further results have been reported by other
authors, including some by members of the SciPost Editorial College. This is thus a timely
submission that would greatly benefit from a prompt publication.

Sincerely, the authors

---

## Round 3 · List of Changes

• Change of the title (from ,,Exact Thermal Properties of Integrable Spin Chains") to ,,Exact Thermal Properties of free-fermionic Spin Chains''

---

## Editorial Decision

published